# Structural insight into an Arl1–ArfGEF complex involved in Golgi recruitment of a GRIP-domain golgin

H. Diessel Duan [1,3], Bhawik K. Jain [2,3], Hua Li[1], Todd R. Graham [2] ✉ & Huilin Li [1] ✉

Arl1 is an Arf-like (Arl) GTP-binding protein that interacts with the guanine nucleotide exchange factor Gea2 to recruit the golgin Imh1 to the Golgi. The Arl1–Gea2 complex also binds and activates the phosphatidylserine flippase Drs2 and these functions may be related, although the underlying molecular mechanism is unclear. Here we report high-resolution cryo-EM structures of the full-length Gea2 and the Arl1–Gea2 complex. Gea2 is a large protein with 1459 residues and is composed of six domains (DCB, HUS, SEC7, HDS1-3). We show that Gea2 assembles a stable dimer via an extensive interface involving hydrophobic and electrostatic interactions in the DCB and HUS region. Contrary to the previous report on a Gea2 homolog in which Arl1 binds to the dimerization surface of the DCB domain, implying a disrupted dimer upon Arl1 binding, we find that Arl1 binds to the outside surface of the Gea2 DCB domain, leaving the Gea2 dimer intact. The interaction between Arl1 and Gea2 involves the classic FWY aromatic residue triad as well as two Arl1-specific residues. We show that key mutations that disrupt the Arl1–Gea2 interaction abrogate Imh1 Golgi association. This work clarifies the Arl1–Gea2 interaction and improves our understanding of molecular events in the membrane trafficking.

Eukaryotic cells are compartmentalized by membranes. To maintain intracellular homeostasis, proteins and lipids are transported by membrane-enclosed small vesicles that bud from donor compartments and fuse to acceptor compartments[1]. The Arf (ADP-ribosylation factor) GTPases including the Arl (ADP-ribosylation factor-like) subfamily are master regulators of membrane trafficking and are involved in all steps of vesicular transport[2,3]. Like the Ras superfamily GTPases, the Arf and Arl proteins such as Arl1 are activated by guanine nucleotide exchange factors (GEFs). Active Arf/Arl1 are in the GTP-bound state and associate with the membrane. The GTPase-activating proteins (GAPs) convert Arf/Arl1 into a GDP-bound inactive state. Inactive Arf/Arl1 then dissociate from the membrane and become cytosolic. Arf proteins recruit coat complexes such as COPI and clathrin adaptors to the Golgi for vesicle budding. For example, Arf1 recruits a coiled-coil tether, Rud3, to the *cis*-Golgi[4], whereas Arl1 recruits the tethering

factor Imh1 to the *trans*-Golgi network along with other tethering proteins required for vesicle docking and fusion[5].

Arf/Arl/Rab GTPases and associated GAPs and GEFs interact to form a complex regulatory network. In budding yeast, the ArfGEF Sec7 is recruited to the Golgi by activated Arf1, Arl1, Ypt1 (Rab1), and Ypt31/32 (Rab11)[6]. Ypt1 is also involved in recruiting the ArfGEF Gea2 to the Golgi where it binds to Arl1 and the phosphatidylserine flippase Drs2[7,8]. Among all Arf family GTPases, Arl1 stands out as operating predominantly at the *trans*-Golgi network (TGN)[9], a central sorting hub for vesicular transport. Interestingly, the guanine nucleotide exchange of Arl1 is not catalyzed by Gea2 and is catalyzed by Syt1 instead. Syt1 is a dual ArfGEF/ArlGEF that contains a conserved PH (pleckstrin homology) domain required for Golgi-targeting[10,11]. In contrast, Gea2 lacks such a PH domain and has been proposed to interact with the membrane via an amphipathic linker[12].

[1]Department of Structural Biology, Van Andel Institute, Grand Rapids, MI, USA. [2]Department of Biological Sciences, Vanderbilt University, Nashville, TN, USA. [3]These authors contributed equally: H. Diessel Duan, Bhawik K. Jain. ✉e-mail: tr.graham@vanderbilt.edu; huilin.li@vai.org

Interestingly, Gea2 recruitment to the Golgi does not require Arl1-GTP binding[8]. Instead, Arl1 binds to Gea2 to form a ternary complex with Drs2, thereby activating the Drs2 flippase activity[8]. Drs2 hydrolyzes ATP to translocate phosphatidylserine against a concentration gradient toward the cytosolic leaflet. The flippase activity leads to membrane remodeling including vesicle budding[8,13–16]. The Arl1-Gea2–Drs2 complex is further involved in TGN recruitment of Imh1, a GRIP-domain golgin tether that helps dock endosome-derived vesicles at the TGN. Therefore, Arl1-GTP plays a dual role in recruiting Imh1—by activating the Drs2 and binding to the GRIP domain. Under this paradigm, Gea2 associates with the membrane to bind and sequester the autoinhibitory Drs2 C-terminal tail, thereby stimulating the Drs2 flippase activity. Phosphatidylserine transport by activated Drs2 from the luminal leaflet to the cytosolic leaflet progressively expands the cytosolic leaflet surface at the expense of the luminal leaflet, leading to membrane bending toward the cytosol. Consistent with this mechanism, deletion of either *DRS2* or *GEA2* compromises membrane asymmetry as well as Golgi localization of Imh1[8].

In this work, to understand the molecular interactions underlying membrane remodeling and trafficking, we determine the cryo-EM structures of the Gea2 dimer alone and its complex with Arl1 using purified *S. cerevisiae* Gea2 and Arl1. These structures allow us to redefine the boundaries of multiple Gea2 domains and to discover that Arl1 binds the Gea2 DCB domain on the opposite side of the previously reported crystal structures of the homologous complex ARL1–BIG1[17,18]. We further perform extensive structure-based mutagenesis to relate the structural insights with cellular functions, and reveal that mutations that disrupt the interaction of Arl1–Gea2 also abolish the Golgi association of Imh1.

## Results and discussion
### Cryo-EM structure determination of the Gea2 dimer and the Arl1–Gea2 complex
We expressed the *S. cerevisiae* Gea2 with an N-terminal 6xHis-tag in *E. coli* and purified the protein with a Co-NTA affinity column followed by size exclusion chromatography (Supplementary Fig. 1a, b). We employed mass photometry to reveal that Gea2 predominantly exists as a dimer in vitro (Supplementary Fig. 2). We determined the cryo-EM structure of the Gea2 dimer at an overall resolution of 3.7 Å (Supplementary Figs. 3, 4, Supplementary Table 1). The three HDS domains (HDS-1, HDS-2, and HDS-3) were partially mobile and had weaker densities in the consensus refined EM map. We next performed masked local refinement, resulting in improved EM densities in these regions.

To study the Arl1–Gea2 interaction, we expressed in *E. coli* an N-terminal 17-residue truncated *S. cerevisiae* Arl1, carrying a Q72L mutation and a C-terminal 6xHis-tag. The Q72L mutation was previously reported to keep the protein in a GTP-restricted state[19], and the removal of the N-terminal 17-residue amphipathic α-helix renders the protein soluble[20]. Both modifications allow Arl1 to adopt the GTP-bound conformation, which is membrane-bound when the amphipathic α-helix is present. We found that the N-terminal truncated Q72L Arl1 readily assembled with Gea2 into a stable Arl1–Gea2 complex[21,22] and demonstrated the in vitro binding of Gea2 with Arl1 by gel filtration chromatography (Supplementary Fig. 1c, d). We determined the cryo-EM structure of the Arl1–Gea2 complex at an overall resolution of 3.3 Å (Supplementary Figs. 5–7, Supplementary Table 1). We also performed masked local refinement to improve the densities of the HDS regions.

We built atomic models for both the Gea2 and the Arl1–Gea2 EM maps, by referencing the AlphaFold2 model of Gea2 and the reported crystal structure of the Arl1[17] and refined these models to good geometry (Supplementary Table 1). Interestingly, Gea2 is predominantly composed of α-helical repeats, with six domains arranged almost linearly from N- to C-terminus, except for the catalytic Sec7 domain that rides above the HUS domain. The structure has allowed us to redefine

the boundaries of the six Gea2 domains, with the N-terminal DCB from 1 – 231 amino acid (aa), the HUS domain from 329 – 536 aa, the catalytic Sec7 domain from 552 to 753, and the three HDS domains: HDS-1 from 799 to 1033, HDS-2 from 1041 to 1208 and HDS-3 from 1218 to the C-terminus (Fig. 1a, b). The domain boundary corrects the previous sequence homology-based definition[23] and complements the recently reported Gea2 composite structure at a lower resolution range of 4.1 to 4.4 Å that was carried out in a different biological context of Arf1 activation[12].

### The Gea2 dimer is not essential to cell viability but is required for Imh1 localization to Golgi
Gea2 is a highly elongated and crescent-shaped molecule, and two Gea2 molecules dimerize via the DCB and HUS domains to perhaps confer stability and structural rigidity (Fig. 1b–e). Two Arl1 molecules bind to the DCB domains on the opposite side of the dimerization interface. We found the structure of Gea2 alone is almost identical to Gea2 in complex with Arl1 (Fig. 1b–e). Therefore, most of our discussion on the Gea2 structure will be based on the higher-resolution Arl1–Gea2 complex structure unless otherwise specified. Gea2 dimerization is primarily driven by hydrophobic interactions at the core DCB region involving Phe-171, Leu-167 and Val-209 (Fig. 2a, b). Phe-171 is not well conserved, but the nearby Arg-174 that could form a strong cation-π interaction with the Phe-171 phenyl ring is highly conserved among the GBF/Gea family (Supplementary Fig. 8). In the homologous BIG/Sec7 proteins, Arg-174 is frequently replaced with leucine at the hydrophobic interface, and BIG1 carrying an L156D mutation at the equivalent position of Arg-174 was found to mislocalize from the Golgi to the cytoplasm[17]. In addition, in the regions flanking the core region, Leu-128 and Val-223 of the DCB domain and Ile-369 of the HUS domain also contribute to the hydrophobic dimer interface (Fig. 2a, b).

Beyond the hydrophobic dimer interface core and at the edges of the dimer interface, electrostatic interaction dominates at the N-terminus of the DCB domain (Fig. 2a, c). Lys-124 in the DCB domain interacts with Gln-364 and is conserved in the GBF/Gea and BIG/Sec7 families (Supplementary Fig. 8). Substitutions including the equivalent Lys-91 in human GBF1 disrupted dimerization, although the mutant protein was still targeted to the Golgi[24]. However, substitution in the equivalent K105 in the human BIG1 (K105D) misdirected the protein to the cytoplasm[17]. There are five additional electrostatic interaction pairs involving Arg-27–Glu460, Thr-24–Glu-464, Arg-80–Ser-467, Lys-17–Thr-414, and Asp-10–Thr-13–Arg-472 (Fig. 2c). The negatively charged residues Glu-460 and Glu-464, the positively charged Arg-472 from the HUS domain are conserved among the GBF/Gea proteins (Supplementary Fig. 9). The positively charged residue Arg-80 is also largely conserved in the GBF/Gea family (Supplementary Fig. 8).

We next performed structure-guided mutagenesis at the dimer interface to interrogate the phenotypic consequences, subcellular localizations, and oligomeric states of Gea2. While deletion of the DCB domain (DCBΔ) caused a loss of viability when expressed in a *gea1Δ gea2Δ* strain, all single and double mutations at the dimer interface supported cell growth (Supplementary Fig. 11). However, the double mutant K124A, L128A failed to dimerize, similar to removal of the entire DCB domain (Supplementary Fig. 12). We further investigated Gea2 mutation on the cellular localization of the GRIP domain protein Imh1, a downstream effector of Arl1. In yeast cells carrying the double mutant Gea2 K124A, L128A that fails to dimerize but is targeted to the Golgi, we found Imh1 was predominantly in the cytosol. The same Imh1 mislocalization effect was observed for the DCB deletion mutant. In contrast, Imh1 was properly localized to the Golgi in the yeast cells carrying the double mutant Gea2 L167A, F171A that remained dimeric as well as in the wild-type Gea2 containing yeast cells (Supplementary Fig. 13a, d). On the other hand, the DCB domain deletion and those double mutations introduced to Gea2 did not affect their localizations to the

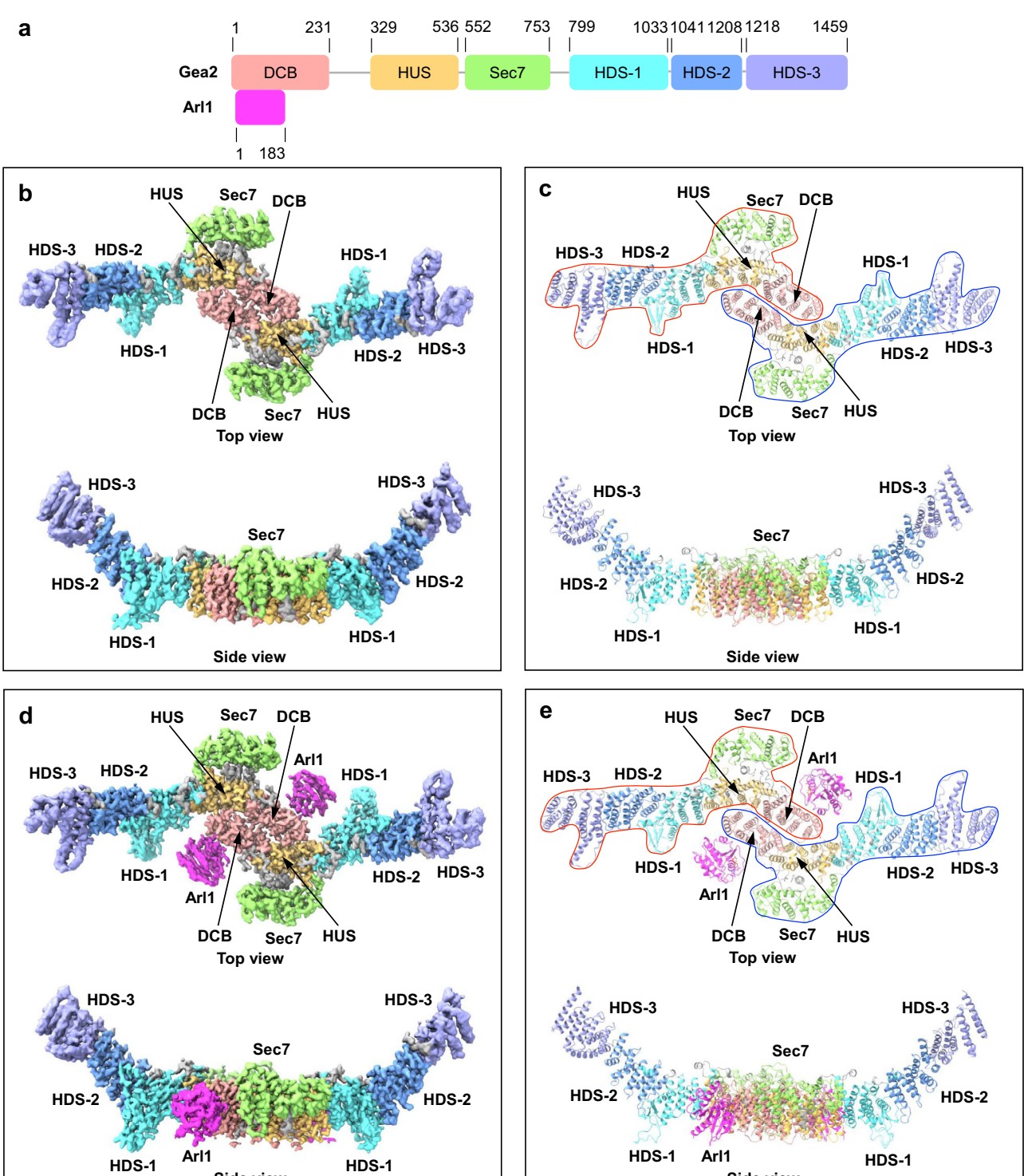

**Fig. 1 | Cryo-EM structures of the Gea2 dimer and Arl1–Gea2 complex. a** Domain architecture of Gea2: DCB (dimerization and cyclophilin binding), HUS (homology upstream of Sec7) and HDS (homology downstream of Sec7). The length of the boxes and lines are scaled to reflect the number of residues. Arl1 is also shown where bound to Gea2. **b, c** Cryo-EM 3D map (**b**) and atomic model (**c**) of the Gea2 dimer with the domains individually colored as in (**a**). **d, e** Cryo-EM 3D map (**d**) and atomic model (**e**) of the Arl1–Gea2 complex in a top and a side view. Gea2 protomers A and B are outlined in a blue and a red shape in the top views.

Golgi (Supplementary Fig. 13b, e). Additionally, we found that Gea2's Golgi localization was unaltered in cells carrying no Arl1, wild-type Arl1 or the Arl1 double mutant L69A, Y78L (Supplementary Fig. 13c, f). These observations suggest that Gea2 dimerization is not essential for cell viability but is required for recruitment of Imh1 to the Golgi. We note that the possibility of the Gea2 double mutant K124A, L128A dimerizing in vivo cannot be ruled out, because a

weakened dimer may not have survived the co-immunoprecipitation conditions.

## The HDS arm is outside the core dimer interface and is partially mobile
The three HDS domains at the C-terminal half are not involved in dimerization, and they form two arms projecting out from the

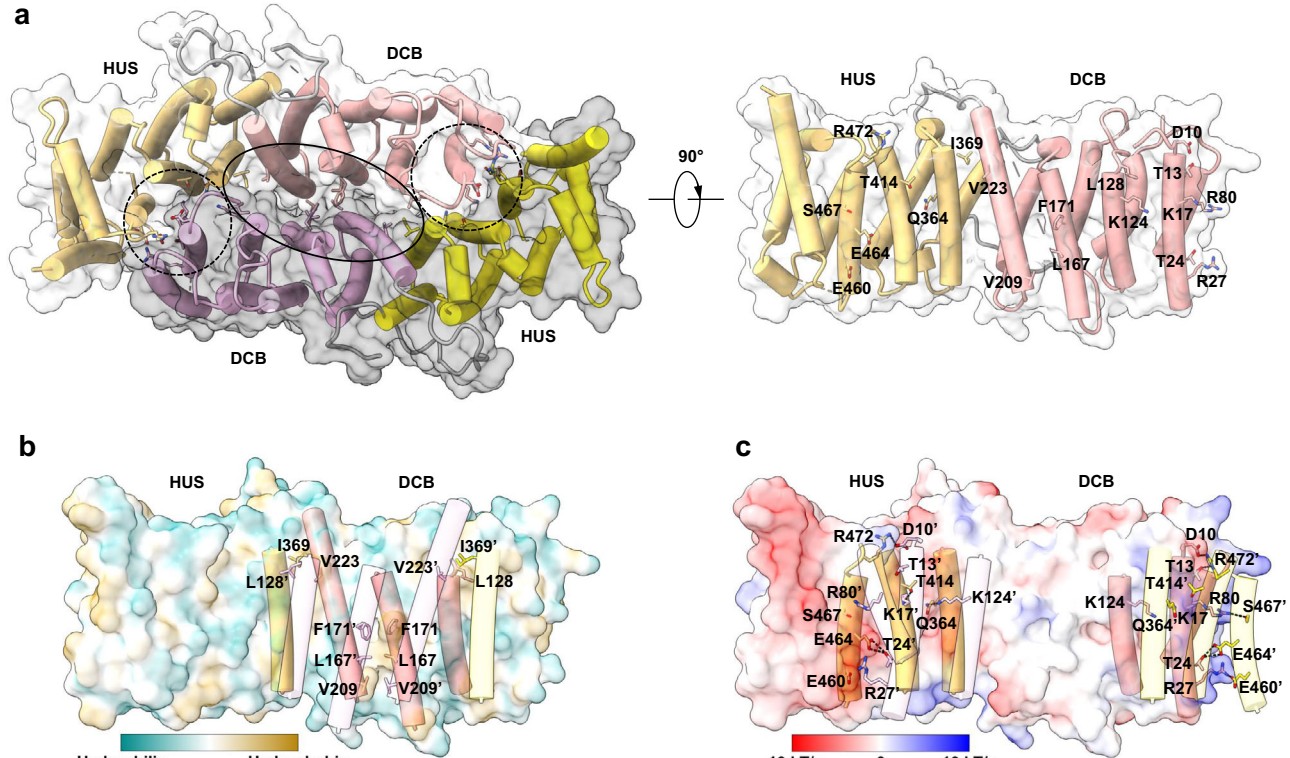

**Fig. 2 | Dimerization interface of Gea2. a** Top (left) and side view (right) of the Gea2 core dimerization region comprised of the DCB and HUS domains, with protomer A surface in grey and protomer B surface in white. The hydrophobic interface region is marked by a black oval, and two electrostatic interface regions are marked by dashed circles. Key residues of promoter B at the dimer interface are shown in sticks and are labeled in the side view. **b** Surface representation of hydrophobicity for protomer B of Gea2 with key residues shown for both protomers in sticks. **c** Surface representation of electrostatic potentials for protomer B of Gea2 with key residues shown for both protomers in sticks. The distance threshold for electrostatic interactions is set to 3.8 Å.

dimerization core (Fig. 1b–e). Apparently, the large number of charges at the edge of the dimer core in the HUS domain act as a hinge to allow arm movement. Indeed, the homodimer of Gea2 is globally asymmetric due to the conformational difference between the two arms of HDS domains. The mobility of the HDS arm is manifested by the fact that protomer B rotates 9° away from the dimer core compared to that of protomer A (Fig. 3a).

### The HUS box is part of a hinge that enables a large-scale tilting of the Sec7 domain

The catalytic Sec7 domain is situated at the junction between the rigid dimer core and the mobile HDS arm (Fig. 1b–e). Sec7 is connected to the HUS domain with an N-terminal linker (termed "N-linker" thereafter) and to the HDS-1 domain with a C-terminal linker (termed "C-linker" thereafter) (Fig. 3b). Superimposition of our Arl1–Gea2 structure with the Arf1–Gea2 complex structure[12] reveals a 35° tilt of the Sec7 domain (Fig. 3c), likely representing a functional transition from the resting-state Gea2 to the nucleotide-free Arf1-bound catalytic intermediate[12]. In contrast, Arl1 binding does not induce Sec7 movement, as the Gea2 structure is essentially the same when alone or bound to Arl1 (Fig. 1b–e). The HUS domain contains a signature HUS box, a highly conserved seven-residue motif of Y/FΦNY/FDCD/E/N (Φ: hydrophobic)[23,25]. The Sec7 N-linker interacts with the HUS box to form a hinge to enable the large-scale Sec7 tilt. The Sec7 hinge is stabilized by a hydrophobic interaction between Cys-486 and Ile-553 and an electrostatic interaction between Asp-485 and Arg-557. We suggest that the previously reported temperature-sensitive phenotype in the single mutants D485G and C486R is due to destabilization of the Sec7 hinge[26]. The Sec7 C-linker is a short α-helix sandwiched between HUS and Sec7. The C-linker is stable and apparently anchors the preceding flexible long loop for the rigid-body motions of the Sec7 domain (Fig. 3b).

### The unexpected binding interface between Arl1 and Gea2

In our cryo-EM structure of the Arl1–Gea2 complex, Arl1 binds to the peripheral surface of the DCB domain. The binding interface involves the N-terminus second and third α-helices of the DCB just outside the dimer core region (Fig. 4a). Arl1 has a Rossmann-like α/β/α core associated by two switch loops (Switch 1 and 2) and an interswitch β-hairpin[27]. The interaction between Arl1 and Gea2 is predominantly hydrophobic and is mediated by Arl1 Switch 2 and the interswitch β-hairpin. Importantly, the interaction involves the classic aromatic triad of Phe-52–Trp-67–Tyr-82, a structural signature of the Arf/Rab family of GTPases for recruiting their respective effectors[28]. The Arl1 aromatic triad (Phe-52–Trp-67–Tyr-82), along with two adjacent hydrophobic residues Val-54 and Leu-69 form extensive hydrophobic interaction with the Gea2 Pro-71, Phe-72, and Leu-103. Pro-71 is highly conserved in the GBF/Gea family. Interestingly, Leu-103 is unique to Gea2 within the GBF/Gea family (Supplementary Fig. 8) but is highly conserved in the BIG/Sec7 family proteins. In addition to the primary interaction mediated by the Arl1 triad, the Switch 2 residues Tyr-78 and Cys-81 also interact with Gea2. Notably, the Arl1 Tyr-78 inserts into the Gea2 hydrophobic cavity formed by Leu-79 and Pro-100. Given that both Tyr-78 and Cys-81 are Arl1-specific residues that are conserved only in Arl1 but not in Arf1 (Supplementary Fig. 10), we speculate that these residues may interact with and specify the Arl1 effectors. In agreement with this possibility, the partnering Gea2 residues Leu-79 and Pro-100 are strictly conserved in the GBF/Gea family (Supplementary Fig. 8).

The Arl1–Gea2 binding interface in our cryo-EM structure is completely different from that of a co-crystal structure of homolog ARL1 bound to a truncated BIG1[17] (Fig. 4b). In fact, Arl1 in our structure binds to the opposite side of the DCB domain of BIG1. Because the Gea2 dimer interface is primarily hydrophobic and highly stable, is four-fold larger than that between Arl1 and Gea2 complex (2524 Å² vs

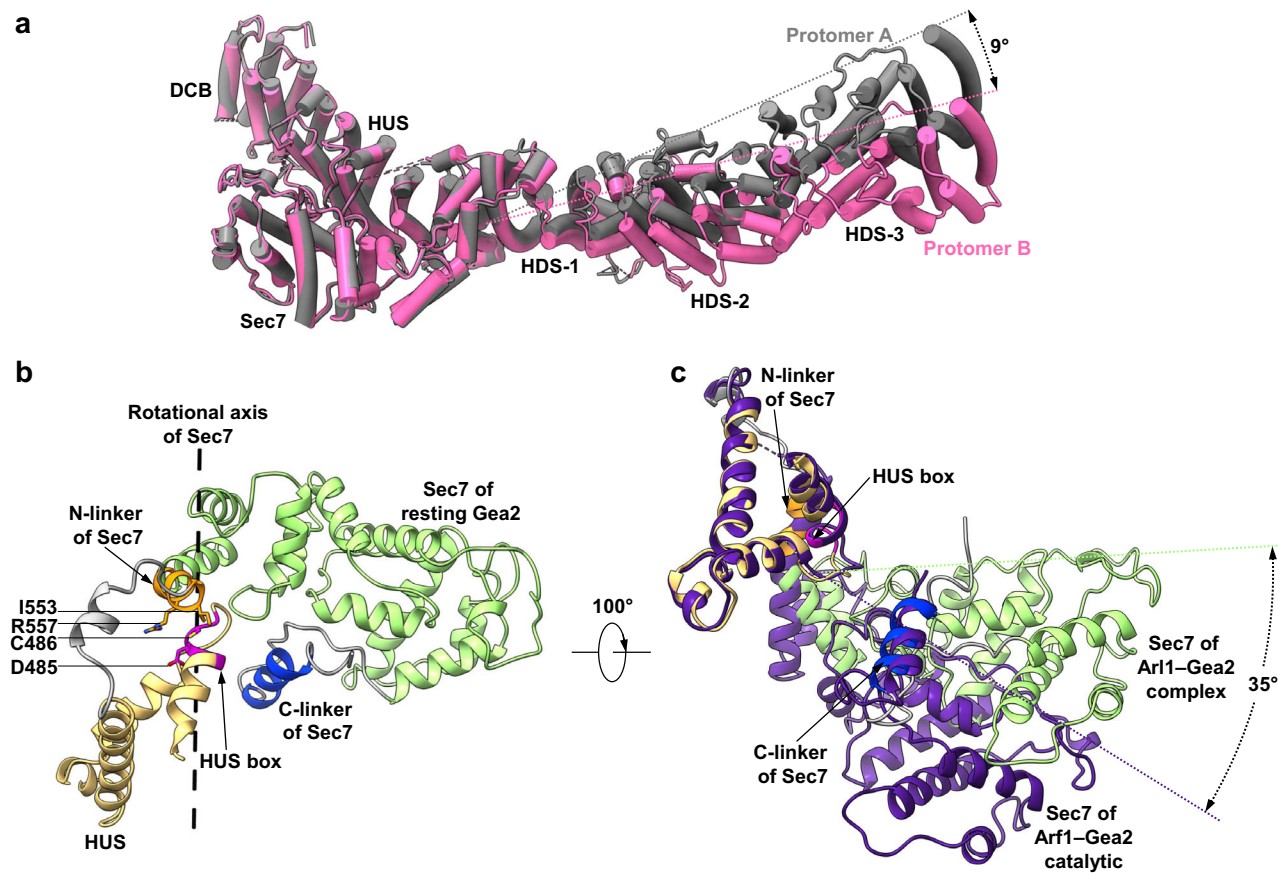

**Fig. 3 | Conformational dynamics of the Gea2 HDS arm and Sec7 domain.**
**a** Superimposition of protomers A (in gray) and B (in pink) based on the rigid DCB and HUS region. The double-headed arrow indicates a 9° tilt between the two HDS arms. **b** The Sec7 domain is connected to the Gea2 main body by three structural elements that may facilitate motions: the HUS box in magenta, the N-linker in orange, and the C-linker in blue. **c** Superimposition of the HUS-Sec7 regions in Arl1–Gea2 (this study) (yellow and green) and Arf1–Gea2 complex (purple; PDB 7URO). The alignment is based on the stationary HUS domain. The Sec7 domain remains in place when bound to Arl1 but swings by 35° hinging around the HUS box during GEF catalysis in the Arf1-bound state.

594 Å²), and involves both DCB and HUS domains, it is highly unlikely that Arl1 can compete off and alter the Gea2 dimer interface. One possible explanation of the difference is that BIG1 truncation abolished the BIG1 dimer, leading to exposure of the hydrophobic DCB dimerization interface and hence the interaction with the also hydrophobic ARL1 binding interface. Superimposition of the DCB domains in these two structures shows that the residues in BIG1 DCB that ARL1 binds are strikingly equivalent to the dimerization interface of the Gea2 DCB, both are predominantly hydrophobic (Fig. 4b). However, a most recent structural study of a thermophilic fungus Sec7, a close homolog of BIG1, revealed that Sec7 dimerizes via its C-terminal HDS-4 domain rather than via the DCB and HUS domains[29]. The HDS-4 domain is unique to this family and is absent in Gea2. Therefore, it is most likely that the BIG1 dimer interface is distinct to that of Gea2, and that the ARL1 binding surface on the BIG observed by co-crystallography is physiologically relevant[17,18]. Their different binding modes are illustrated by a side-by-side comparison of the Arl1–Gea2 and putative ARL1–BIG1 domain architectures (Fig. 4c, d). The difference may be attributed to the fact that Gea2 and BIG1 belong to different subfamilies of ArfGEF.

Consistent with the Arl1–Gea2 interface we observed in this study, ARL1 uses highly similar binding surface to interact with its other downstream effectors such as Golgin-245 and Arfaptin-2 despite the difference in the orientations of the two interacting helices from the individual effectors (Supplementary Fig. 14). Because ARL1–Golgin-245 and Arl1–Imh1 are conserved, if the binding pattern in ARL1 holds true

in Arl1, Arl1 likely needs to dissociate from the Gea2 dimer and be activated again to a GTP-bound state before it can interact with the Imh1 GRIP domain. Alternatively, another pool of Arl1 could be involved in recruiting Imh1 to the Golgi. Nonetheless, these data make it unlikely that our solved Arl1–Gea2 complex can also interact with the GRIP domain of Imh1, because Arl1 is unable to interact with both Gea2 and Imh1 simultaneously due to the fact that the same binding surface is involved in both interactions.

We next performed structure-guided mutagenesis to investigate the cellular effect of altering the Arl1–Gea2 interface, with an emphasis on mutating the conserved residues that are specific to Arl1 and Gea2, respectively. We found that the Gea2 P71A and L79A variants still bound to the wild-type Arl1, but the Arl1 L69A and Y78L variants showed reduced interactions with the wild-type Gea2 in our protein-protein interaction assay (Fig. 5). The Arl1 Leu-69 is from the Arl1 Interswitch hairpin and is proximal to the conserved Phe-Trp-Tyr aromatic triad. This residue likely plays an integral role in stabilizing the hydrophobic interaction with the Gea2. Likewise, the bulkiness of the Arl1-specific Tyr-78 is also required for its contribution to the hydrophobic interaction with Gea2. Indeed, the double mutant Arl1 L69A, Y78L completely abolished the Arl1–Gea2 interaction (Fig. 5). In a negative control, we showed that deletion of the essential DCB domain also abolished Gea2's interaction with Arl1 (Fig. 5). Using a late Golgi marker Sec7-GFP, we further showed that while the Arl1 mutant C81A was properly colocalized to the Golgi like the wild-type Arl1, localizations of Arl1 single mutants L69A, Y78L and double mutant

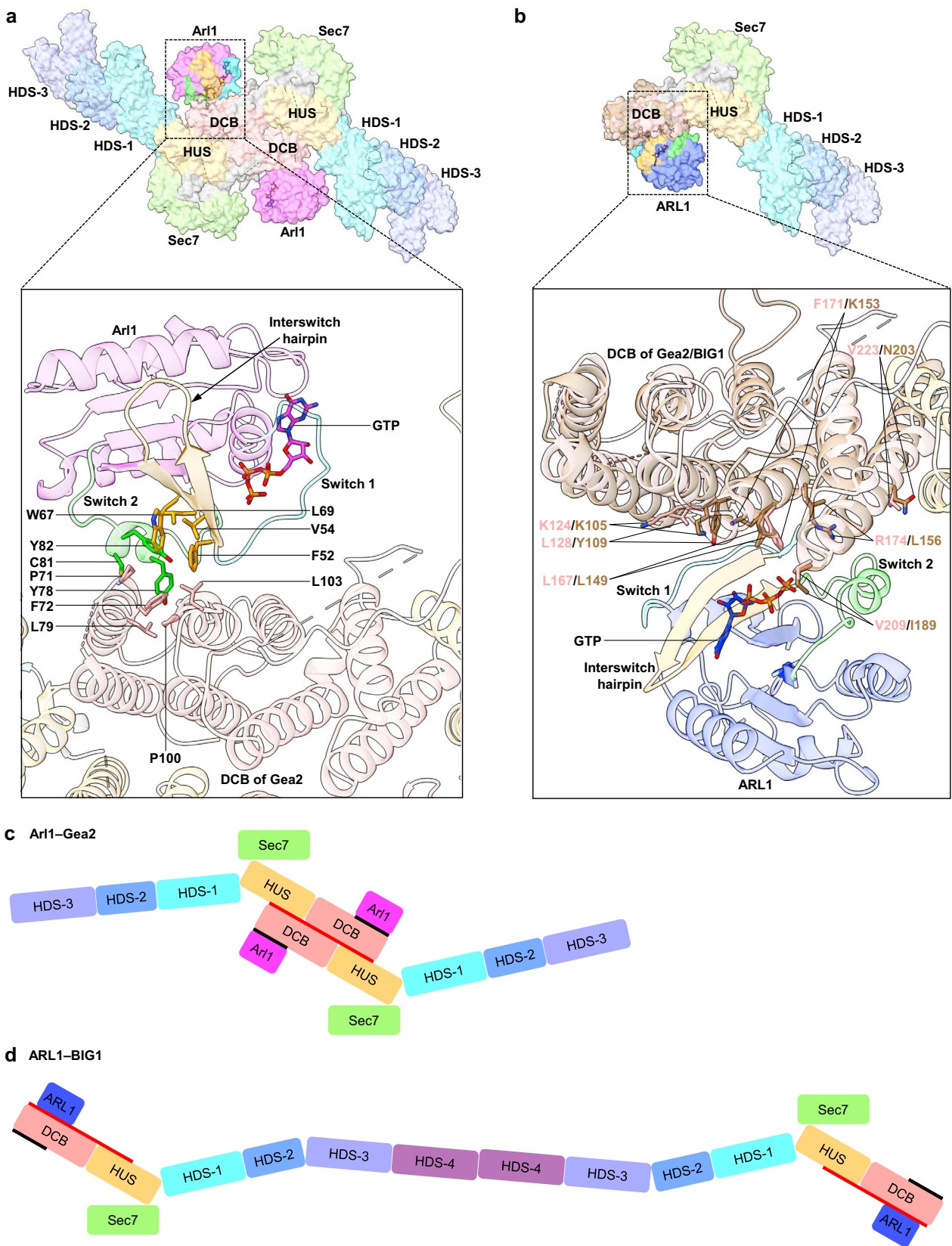

L69A, Y78L were all compromised. Consequently, mCherry tagged Imh1 was mislocalized to the cytoplasm in the yeast cells carrying the Arl1 L69A or Y78L single mutation or L69A, Y78L double mutation, signifying the importance of the Arl1 binding surface for downstream effector targeting (Fig. 6). Intriguingly, these Arl1 mutants showed a diffuse cytoplasmic distribution, suggesting that they are primarily

GDP-bound. Both L69 and Y78 are far away from the γ-phosphate of the bound GTP in our structure, and their mutations are unlikely to directly affect GTP binding. However, these mutations may perturb nucleotide exchange by Syt1[10,11] if the Arl1 surface that binds Gea2 also binds Syt1. Alternatively, the Arl1–Gea2 interaction may stabilize Arl1-GTP by protecting it from Gcs1, the Arl1 GAP[30]. Thus, mutations that

**Fig. 4 | Interface between Arl1 and Gea2 in comparison with interface between ARL1 and BIG1. a** Top panel shows a bottom surface view of the Arl1–Gea2 complex structure obtained by cryo-EM in this study. Bottom panel is a zoomed view of the area in the dashed square in the top panel, showing the interface residues in sticks of Arl1 and the DCB domain of Gea2 protomer A. **b** Top panel is a surface view of superimposition of the DCB domain (brown) in the crystal structure of ARL1–BIG1 DCB (PDB 5EE5) with homolog DCB domain (pink) in the Arl1–Gea2 structure, showing that ARL1 binds to the lower surface of BIG1 DCB which is involved in dimerization in Gea2. Only protomer A is shown for clarity. Bottom panel is a zoomed cartoon view of the region in the dashed square in the top panel. ARL1 binds to the lower DCB surface in BIG1; the corresponding DCB surface in Gea2 is involved in dimerization. Key residues at the lower surface of the two DCB domains are shown in sticks, and they are largely conserved. **c**, **d** Comparison of the domain architectures of Arl1–Gea2 (**c**) and the putative ARL1–BIG1 complex (**d**), both shown in a top view. The red and black lines mark the interior and the exterior surface of DCB-HUS domains, respectively, as in the Arl1–Gea2 structure.

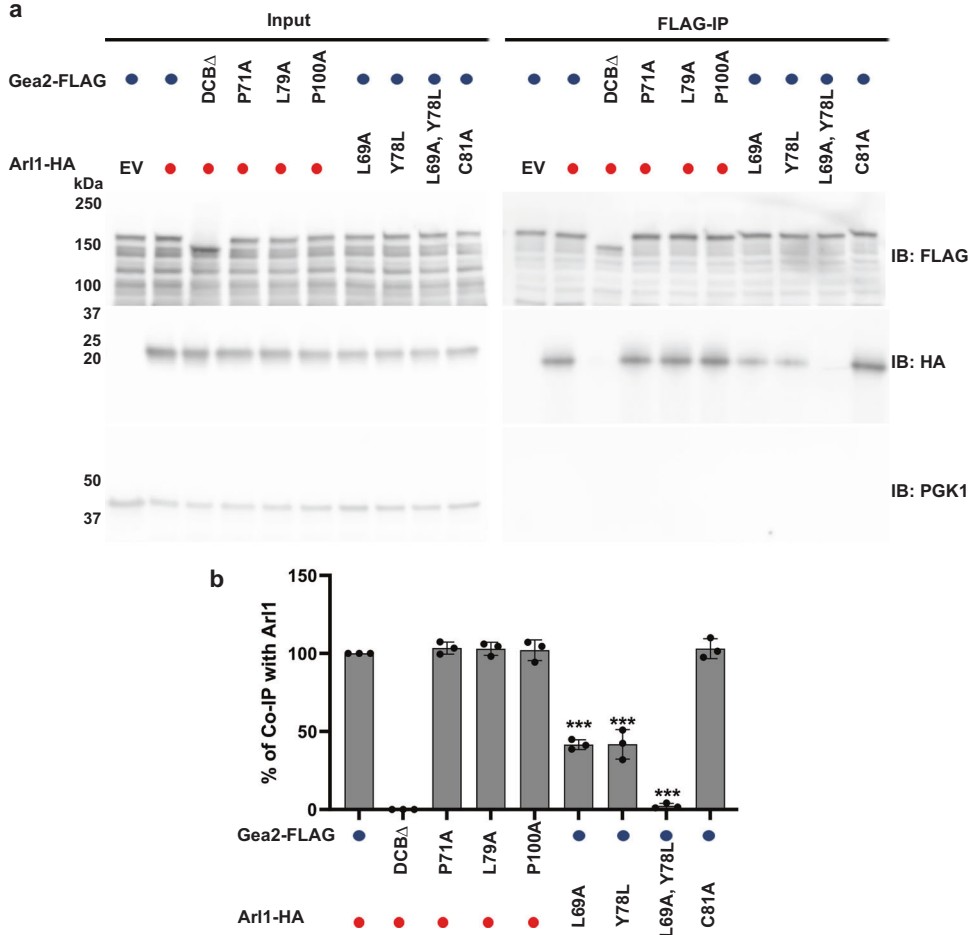

**Fig. 5 | Mutations at the interface between Arl1 and Gea2 affect their interaction. a** As a control, deletion of Gea2 DCB domain abolished Arl1–Gea2 interaction. Individually mutating the Arl1-specific residues L69 and Y78 significantly reduced Arl1 binding to Gea2 and Arl1 L69A, Y78L double mutation abolished Arl1–Gea2 interaction. 3xFLAG tagged WT Gea2 or Gea2 mutant variants were co-expressed with 3xHA tagged WT Arl1 or Arl1 mutant variants in *gea2Δarl1Δ* cells. Gea2-3xFLAG was immunoprecipitated by anti-FLAG M2 magnetic beads, and the coprecipitated Arl1-3xHA was detected with anti-HA antibody. **b** Quantification of coprecipitated Arl1 ($n = 3$). Comparisons were calculated via a One-Way ANOVA followed by Dunnett's multiple comparisons test. $n = 3$ represents data from three independent biological replicates. Error bars represent SD. *** indicates $p$ value < 0.001. Source data are provided as a Source Data file.

disrupt the Arl1–Gea2 interaction could promote Arl1-GTP to Arl1-GDP conversion by Gcs1, and loss of membrane association.

## Membrane topology of the Arl1–Gea2 complex

At first glance, one would assume that the concave surface of the Arl1–Gea2 complex may bind and bend the membrane, much like arfaptins that are an effector of ARL1 and induce membrane bending through their BAR domains[31]. Nonetheless, three lines of evidence argue against such a model for Gea2. First, superimposition of the AlphaFold2 model of an intact Arl1 with the N-terminal truncated Arl1 in our Arl1–Gea2 structure showed that the N-terminal amphipathic α-helix points to the opposite side of the concave surface of Gea2 (Fig. 7a). Second, the heel-like amphipathic motif in the Gea2 HDS-1 domain that likely embeds in the membrane[12] also points to the

opposite side of the concave surface (Fig. 7a). Third, the Gea2 concave surface is hydrophilic and suitable for solvent exposure, but the opposing flat surface is positively charged and compatible with binding to the negatively charged membrane surface (Fig. 7b). Therefore, we conclude that the flat surface rather than the concave surface of the Arl1–Gea2 dimer docks with the Golgi membrane. Interestingly, the Gea2 heel-like motif is highly dynamic in solution; it is invisible in the Gea2 alone structure and becomes visible only in the Arl1–Gea2 complex. This observation suggests that Arl1 promotes Gea2's association with the membrane. In the Arl1–Gea2 structure, a total of four structural elements, two Arl1 N-terminal α-helices and two Gea2 heel-like motifs, are inserted into the membrane, perhaps to prolong their residence time in the TGN for membrane remodeling. Given that Gea2 is an Arf1 guanine nucleotide exchange factor, we next examined by

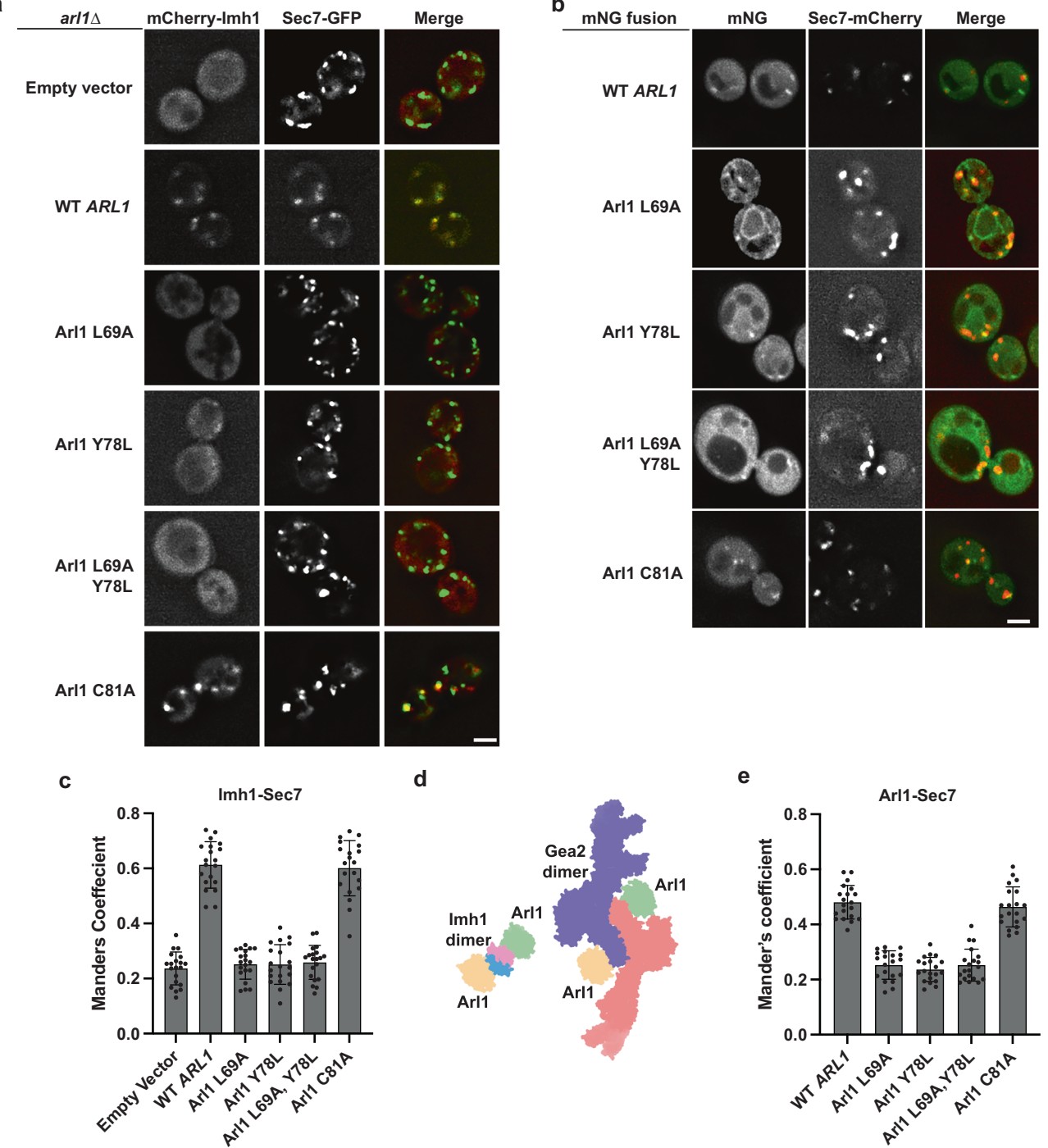

**Fig. 6 | Mutations in the binding surface of Arl1 for Gea2 affect cellular localization of Imh1. a** Localization of mCherry-Imh1 in *arl1Δ* cells expressing empty vector (pRS313), Arl1 WT or Arl1 mutants. **b** Localization of mNeonGreen (mNG) tagged WT Arl1 or Arl1 mutants. Scale bar is 2 μm. Quantification via Manders' Coefficient of the proportion of mCherry-Imh1 (**c**) or mNG tagged Arl1 (**e**) colocalized with a late Golgi marker Sec7. For all quantifications, data from (*n* = 20) cells from three independent experiments of biological replicates were obtained and analyzed. Comparisons were calculated via a One-Way ANOVA followed by Tukey's post hoc test. Error bars represent SD. **d** Illustration of Arl1 at TGN binding to Imh1 (PDB 1UPT) and to Gea2 using same surface region. The two structures are aligned by the lower left Arl1 and viewed from the membrane. Detailed interfaces are shown in Fig. 4a and Supplementary Fig. 14a. Source data are provided as a Source Data file.

computational docking if Gea2 could bind Arl1 and Arf1 simultaneously. Our docking results show that Gea2 can bind Arl1-GTP and Arf1-GDP at the same time without physical conflict (Fig. 7c). However, the following nucleotide exchange reaction by Gea2 will swing the catalytic Sec7 domain by 35° toward the membrane, thereby pushing the Arf1-NF (nucleotide-free) halfway into the hydrophobic core of the membrane (Fig. 7d). Further studies are needed to understand if Gea2 can act on both Arf1 and Arl1 simultaneously.

## Summary and perspective
We have capitalized on the constitutively active mutant Arl1 locked in the GTP-bound state to capture the interaction between Arl1 and the

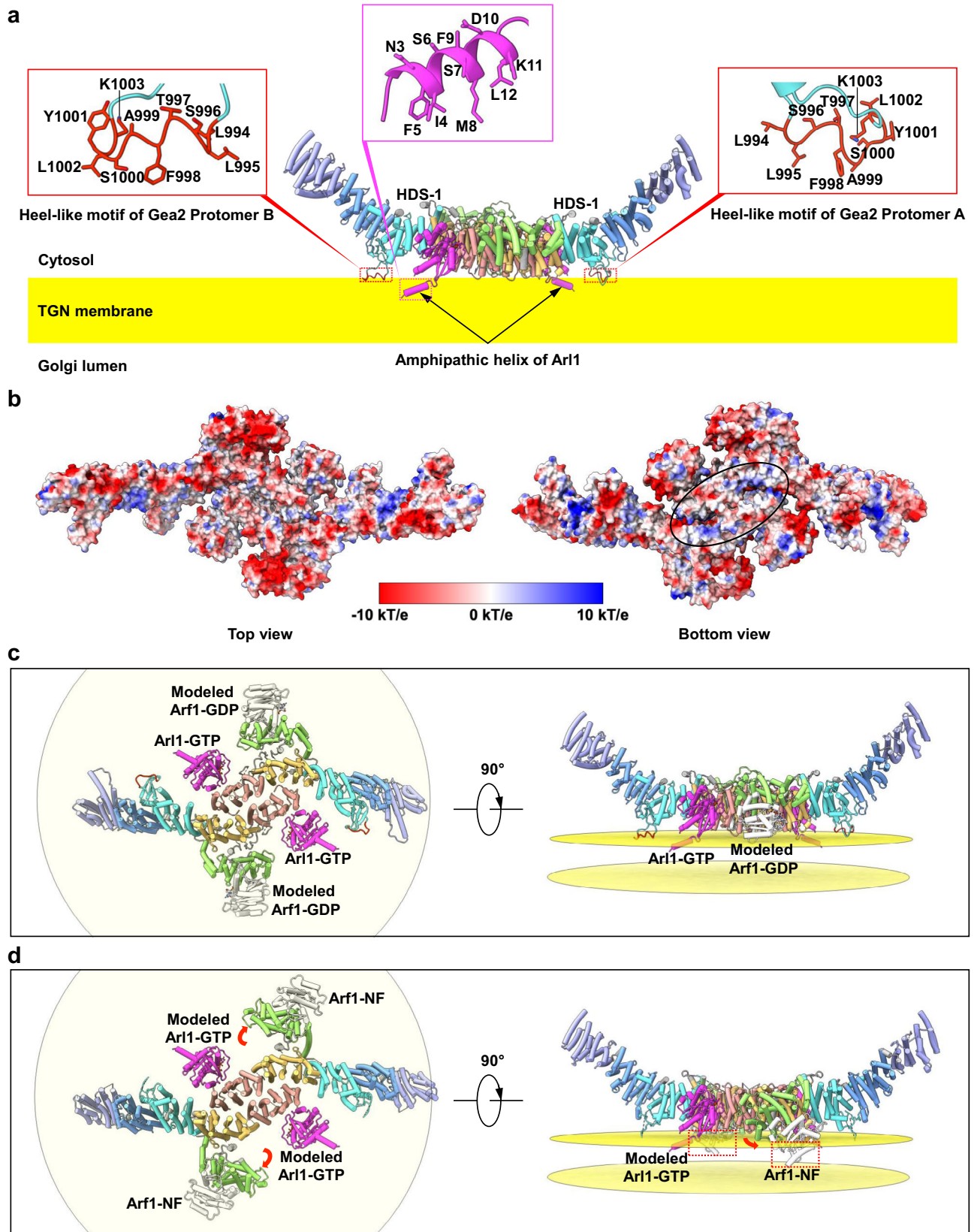

full-length Gea2 dimer. Our study revealed the extensive hydrophobic dimerization interface, implying that Gea2 functions as a dimer in vivo. We showed that Arl1 binds to the external surface of the N-terminal DCB domain outside the hydrophobic core of the Gea2 dimer, and this interface is entirely different from the interface observed in the crystal structures of the truncated homolog ARL1–BIG1 complex. We

discovered that the Arl1-specific residues Leu-69 and Tyr-78 at the Arl1–Gea2 interface are essential for the recruitment of the golgin protein Imh1. Our data support a model whereby the Arl1–Gea2 complex docks onto the Golgi membrane with its flat surface at the opposite side of the concave surface[12]. The Arl1–Gea2 structure has brought us one step closer toward solving the bigger puzzle of how the

**Fig. 7 | Membrane topology of the Arl1–Gea2 complex. a** A plausible docking pose of the Arl1–Gea2 complex on the Golgi membrane. Arl1 is in magenta cartoon. The Arl1 N-terminal amphipathic α-helix is truncated in our structure and is modeled based on AlphaFold2 prediction (AF-P38116-F1-model_v3). The Gea2 heel-like motif is in red cartoon. Zoomed views of the Arl1 amphipathic helix and the Gea2 heel motif are shown above the Arl1–Gea2 cartoon structure with their respective residues shown in sticks and labeled. **b** Electrostatic potential maps of the Arl1–Gea2 complex in a top and a bottom view. The most likely surface patch for interactions with the negatively charged phospholipids of the membrane is demarcated by a black oval in the bottom view. **c** Bottom and side views of modeled Arf1-GDP binding (PDB 1R8S) to the current Arl1–Gea2 complex. There is no physical conflict, suggesting that Gea2 may bind both Arl1-GTP (magenta) and Arf1-GDP (white) simultaneously. **d** Bottom and side views of modeled Arl1-GTP binding to the Arf1–Gea2 complex (PDB 7URO). The red arrows indicate the rotation of Sec7 domain from Arf1-GDP bound state to the Arf1-NF bound state. Note that Arf1-NF (white) is halfway sunken into the TGN lipid bilayer (boxed in red), which is presented by two parallel yellow planes in both (**c**) and (**d**).

Arl1–Gea2 complex stimulates the Drs2 flippase activity. Gea2 regulation of the P4-ATPase Drs2 flippase is an emerging function for an ArfGEF and is an under-studied topic in the field of membrane remodeling and trafficking.

## Methods

### Molecular cloning

The gene sequences encoding *S. cerevisiae* Arl1 and Gea2 were PCR-amplified from genomic DNA of *S. cerevisiae* BY4741 obtained from Horizon Discovery (Waterbeach, UK) with pairs of primers 5′-GTC TCTCCCATGGGTAACATTTTTAGTTCAATGTTTG-3′/5′-GGTTCTCCCC AGCTAACTGTTCCTCTTTTATAA-3′ and 5′-TACTTCCAATCCAATGC AATGAGTGATAGGGAATTCGTC-3′/5′-TTATCCACTTCCAATGCCTTA ATCCTTTTCTACATCAGATAACTTC-3′, respectively. The genes were inserted using ligation-independent cloning into vectors acquired from the DNASU Plasmid Repository (Tempe, AZ), pMCSG28 for the expression of Arl1 and pMCSG21 for the expression of Gea2. Using Q5 site-directed mutagenesis kit (New England Biolabs, Ipswich, MA), expression vectors for Q72L Arl1 and also an N-terminal 17-residue truncated Q72L Arl1 (termed "ΔNT17 Q72L Arl1") were also generated. After DNA sequence verification (GENEWIZ, South Plainfield, NJ), the *E. coli* strain NiCo21(DE3) (New England Biolabs, Ipswich, MA) was transformed with constructed vectors. To overcome codon bias, an additional plasmid pRARE2 isolated from *E. coli* strain Rosetta 2 (Novagen, Madison, Wisconsin) was also included in the transformation for Gea2 expression. The expressed protein Arl1 had a C-terminal 6xHis-tag, and Gea2 had an N-terminal 6xHis-tag to facilitate their purifications individually.

### Protein expression and purification

Cells were grown in 2 liters of Terrific Broth supplemented with 2 mM MgSO₄ along with carbenicillin (100 μg/mL), spectinomycin (100 μg/mL), and chloramphenicol (50 μg/mL) when appropriate at 37 °C, shaking at 250 rpm, to an $OD_{600}$ of ~0.7. For Gea2 expression, 0.5% glucose was also included to suppress basal expression. After fully cooling the cultures to 18 °C, gene expression was induced with 0.1 mM IPTG, and cells were grown for an additional 16 h at this lower temperature. Cells were harvested by centrifugation at 11,899 × *g* at 4 °C for 6 min, cell pellet was washed and resuspended in protein buffer (20 mM HEPES, pH 7.4, 150 mM NaCl and 1 mM MgCl₂), then dripped into liquid nitrogen and finally stored at −80 °C before cell lysis. Cells were lysed using a SPEX Freezer/Mill 6875 (SPEX SamplePrep, Metuchen, NJ) with the following settings: pre-cool 1 min, run time 2 min, cool time 1 min, cycles 5, and rate 15 CPS (cycle per second). The milling was repeated one time, then the lysed cells were resuspended in buffer also containing 1 mM AEBSF (Chem-Impex International, Wood Dale, IL) and 10 μL Benzonase Nuclease HC (EMD Millipore, Burlington, MA). Additionally, 1 mM GTP (Chem-Impex International, Wood Dale, IL) was also included for purification of ΔNT17 Q72L Arl1. The resuspension was further incubated at 4 °C for 30 min with gentle stirring. After centrifugation at 24,000 × *g* for 30 min at 4 °C, the supernatant was filtered through a 0.22-μm syringe filter. The resulting protein solution was mixed with 5 mL pre-equilibrated TALON Metal Affinity Resin (Takara Bio USA, San Jose, CA) and incubated at 4 °C for 3 h with stirring. The mixture was then transferred to an Econo-Column chromatography column (Bio-Rad Laboratories, Hercules, CA) at 4 °C. After flow-through collection, the column was washed with wash buffer (20 mM HEPES, pH 7.4, 500 mM NaCl and 1 mM MgCl₂) containing 0 mM and 10 mM imidazole in sequence in 20 bed volumes. Finally, the column was developed with 3 bed volumes of wash buffer containing 150 mM imidazole, and the eluate was collected in 0.5 mL fractions. After estimation of protein concentrations with a Nanodrop spectrophotometer (Thermo Fisher Scientific, Waltham, MA), fractions with concentration greater than 1 mg/mL were pooled and concentrated to 0.5 mL using a 10 kDa or 50 kDa cutoff filter (EMD Millipore, Burlington, MA) when appropriate. Finally, the protein concentrate was further purified with a Superose 6 Increase 10/300 GL column (Cytiva, Marlborough, MA) at a flow rate of 0.5 mL/min in protein buffer (20 mM HEPES, pH 7.4, 150 mM NaCl and 1 mM MgCl₂). The peak fractions of target proteins were pooled and concentrated before prompt use or flash-freezing in liquid nitrogen and storage at −80 °C.

### Mass photometry

Mass photometry analysis was performed using 44.2 nM Gea2 on a Refeyn TwoMP mass photometer (Refeyn Ltd, Oxford, England), calibrated with β-amylase and thyroglobulin to generate a standard curve. Gaussian distributions were fitted to the peaks to calculate the average molecular mass and standard deviation.

### In vitro binding analysis by gel filtration

Gea2 in 0.74 μM monomer concentration was incubated with 10-fold molar excess of ΔNT17 Q72L Arl1 in protein buffer supplemented with 1 mM TCEP and 1 mM GTP at 4 °C overnight. The Arl1–Gea2 complex was purified with a Superose 6 Increase 10/300 GL column (Cytiva, Marlborough, MA) at a flow rate of 0.5 mL/min in protein buffer supplemented with 1 mM TCEP. Peak fractions were then analyzed by SDS-PAGE.

### Cryo-EM grid preparation and data collection

The assembly of Gea2 and Arl1 complexes was achieved by overnight incubation at 4 °C of the mixture of Gea2 (7.5 mg/mL) and ΔNT17 Q72L Arl1 (2.7 mg/mL) with a monomer molar ratio of 1:3 in the protein buffer with additional 1 mM TCEP and 1 mM GTP. After glow discharge cleaning for 30 s in a Gatan Solarus Model 950 plasma cleaner (Gatan, Pleasanton, CA), the 300-mesh gold Quantifoil R2/1 holey carbon grids (Structure Probe, West Chester, PA) were applied with 3 μL protein sample supplemented with 0.5 mM fluorinated octyl maltoside (Anatrace, Maumee, OH) beforehand for enhancement of particle distribution. Sample vitrification was carried out in a Vitrobot Mark IV (Thermo Fisher Scientific, Waltham, MA) at 5 °C and 100% relative humidity with the following settings: blot time 2.5 s, blot force 2, wait time 0 s. The EM grids were plunge-frozen in liquid ethane, cooled by liquid nitrogen. Automated cryo-EM data collection was performed on a 300 kV Titian Krios electron microscope controlled by SerialEM v4.0.12[32] in multi-hole mode. Movie stacks were recorded at 105,000× magnification with the objective lens defocus values varied from −1.3 to −1.7 μm, in a K3 direct electron detector (Gatan, Pleasanton, CA) in super-resolution mode. The calibrated image pixel size was 0.828 Å at

specimen level. A GIF Quantum energy filter was also used to remove inelastically scattered electrons with the slit width set to 20 eV. For each movie micrograph, a total of 75 frames were captured with 1.5 s exposure time and a total electron dose of 69 $e^-$/Å$^2$.

## Cryo-EM image processing and 3D reconstruction
Image processing was carried out in cryoSPARC v3.3[33]. Movie stacks were aligned by patch motion correction for beam-induced movement, then the contrast transfer function (CTF) parameter of each micrograph was determined by patch CTF estimation, and the CTF effect was subsequently corrected. After manual inspection to remove poor-quality images, the remaining micrographs were frame aligned and dose weighted. Raw particle images were picked using templates generated from a preliminary 3D map. The picked raw particles were subjected to several rounds of 2D classification. After selecting good classes and removing duplicates with a minimum separation distance of 50 Å, the resulting particles were used for ab initio 3D reconstructions. After two rounds of heterogeneous refinement, the good class particles were further processed by non-uniform refinement to obtain the consensus map. Local refinement with a soft mask covering the C-terminal 565-residue region in Gea2 was also performed to improve the density in the HDS arms. The consensus and local maps were postprocessed by DeepEMhancer[34]. Finally, the composite map was calculated in UCSF Chimera v1.16[35] by adding back the new locally refined volume after its subtraction from the consensus map.

## Model building, refinement, and validation
Cryo-EM maps were postprocessed with DeepEMhancer for better visualization of density details. The initial atomic models were built by fitting the Gea2 AlphaFold model[36] and crystal structure of ARL1 (PDB 5EE5) into the corresponding EM maps. The atomic models were then rebuilt manually and refined with *Coot* v0.9.8.3[37] followed by iterative real-space refinement in Phenix v1.20.1[38]. The final models were validated by MolProbity v4.5.1[39]. Protein domain dynamics was analyzed by DynDom[40]. Structural figures were prepared in UCSF ChimeraX v1.4[41].

## Yeast strains, plasmids, and antibodies
The detailed information on the yeast strains and plasmids used in this study is listed in Supplementary Table 2. All yeast culture reagents were purchased from Sigma-Aldrich and BD Biosciences. Yeast strain deletion or chromosomal tagging was performed using PCR amplification and homologous recombination[42]. Yeast strains were grown in YPD or minimal selective media. Yeast transformation was performed using the standard LiAc-PEG method. DNA constructs were created using standard cloning methods and mutation were created by QuickChange mutagenesis using PfuTurbo DNA polymerase from Agilent. ANTI-FLAG M2 antibody produced in mouse (F3165, 1:3500) and Anti-HA antibody produced in rabbit (H6908, 1:1000) were purchased from MilliporeSigma (St. Louis, MO). Anti-Mouse IgG, HRP conjugate (W4021, 1:10,000) and Anti-Rabbit IgG, HRP conjugate (W4011, 1:10,000) were purchased from Promega (Madison, WI). Mouse PGK1 monoclonal antibody (clone 22C5D8, 1:5000) was obtained from Molecular Probes (Eugene, OR).

## Plasmid shuffling assay
Plasmid shuffling assay was performed by spotting 1 OD cells and further 10-fold serial dilutions of the cells onto synthetic defined media (SD-Leu) and SD-5-FOA plates and incubating at 30 °C for 2 days to observe growth.

## Fluorescence microscopy
Cells were grown in synthetic dropout media to mid-log phase at 30 °C. Cells were washed for 3 times and resuspended in fresh SD media and imaged at room temperature on glass coverslip. Images were captured using DeltaVision Elite Imaging System equipped with a 100×, 1.4 NA oil immersion objective lens. Images were deconvoluted using softWoRx software (GE Healthcare Cytiva, Pittsburgh, PA). Images were analyzed in FIJI (FIJI Is Just ImageJ) and colocalization was quantified through the JaCOP (Just another Colocalization Plugin) add-on by computation of the Manders' coefficient in JaCOP as the fraction of overlap between Sec7 and Imh1, Sec7 and Gea2, and Sec7 and Arl1.

## Protein interaction analysis
Yeast cells were grown to mid-log phase in synthetic dropout medium. 50 OD of the cells were harvested and washed with ice cold sterile water. Cells were resuspended in 300 μL lysis buffer (20 mM HEPES, pH 7.2, 100 mM KCl, 5 mM MgCl$_2$, 1% Triton X-100, 5 mM EDTA, 10% glycerol, and complete protease inhibitor tablet) and then lysed using 0.5 mm diameter glass beads using a Disruptor Genie (Scientific Industries) at 4 °C for 10 min at 3000 rpm. The cell lysates were centrifuged at 13,000 rpm for 15 min at 4 °C to clear the lysate. The supernatant was incubated with 20 μL ANTI-FLAG M2 Magnetic Beads (M8823) overnight at 4 °C. The resin was washed three times with washing buffer (20 mM HEPES, pH 7.2, 100 mM KCl, 5 mM MgCl$_2$, 5 mM EDTA, 0.5% Triton X-100) and eluted by adding 30 μL 2X sample buffer and incubating at 95 °C for 5 min. 1% input and 10% of immunoprecipitated samples were resolved by 4–20% gradient SDS–polyacrylamide gel electrophoresis and immunoblotted for specific antibody. The membranes were imaged with AI600 Chemiluminescent Imager (GE Life Sciences). The intensities were determined by ImageJ Fiji software and normalized to the loading control. The data are presented as the mean ± SD of independent experiments *** $p < 0.001$; N.S. not significant; one-way ANOVA with Dunnett's post-hoc multiple comparison test.

## Reporting summary
Further information on research design is available in the Nature Portfolio Reporting Summary linked to this article.

## Data availability
The cryo-EM 3D maps of the *S. cerevisiae* full-length Gea2 and Arl1–Gea2 complex have been deposited in the Electron Microscopy Data Bank under accession codes EMD-28748 and EMD-28743, respectively. The consensus refined EM maps of the Gea2 and Arl1–Gea2 complex are under accession codes EMD-28749 and EMD-28744, respectively. The focused refined maps of the HDS1-3 of protomers A and B in the Arl1–Gea2 complex have accession codes of EMD-28747 and EMD-28746, respectively. The focused refined maps of the HDS1-3 domains of Gea2 protomers A and B in the Gea2 alone sample have accession codes EMD-28750 and EMD-28751, respectively. The atomic models of Gea2 and Arl1–Gea2 have been deposited in the Protein Data Bank under accession codes 8EZQ and 8EZJ, respectively. The EM data are available from the authors upon reasonable request. Source data are provided with this paper.

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

## Acknowledgements

Cryo-EM images were collected in the David Van Andel Advanced Cryo-Electron Microscopy Suite at Van Andel Institute. We thank G. Zhao and X. Meng for facilitating data collection. This work was supported by the U.S. National Institutes of Health grants R35GM144123 (to T.R.G.) and R01CA231466 (to Huilin Li) and the Van Andel Institute (to Huilin Li). Materials generated in this study are freely available upon request to the corresponding authors.

## Author contributions

H.D.D., T.R.G., and Huilin Li conceived and designed the experiments. H.D.D. generated the *E. coli* expression constructs for Arl1 and Gea2 and expressed and purified these proteins. B.K.J. generated the yeast Gea2 and Arl1 mutant constructs and performed cellular studies. H.D.D. performed cryo-EM, image analysis, and 3D reconstruction. H.D.D. and Hua Li built atomic models. H.D.D. wrote the first version of the manuscript. All authors analyzed the data and edited the manuscript.

## Competing interests

The authors declare no competing interests.
