## [Peer Review File · Nature Communications]

Structural insight into an Arl1-ArfGEF complex involved in Golgi recruitment of a GRIP-domain golginREVIEWER COMMENTS

Reviewer #1 (Remarks to the Author):

In this manuscript, the authors provide structure-based evidence to gain insight into the interaction between the late Golgi Arf-GEF Gea2 and the Arf-like small GTPase Arl1. The authors perform cryo-EM showing that Arl1 binds to the outside surface of the dimerization region of Gea2 (DCB domain) without affecting the dimer structure of Gea2. Recently, Fromme lab reported that Arf1 binds to the Sec7 domain of Gea2 (GEF domain) to cause a conformational change of Gea2 and subsequent activation (Muccini et al. 2022, Cell Reports 40:111282). These results support the idea of how the Arf-GEF Gea2 is involved in the function of the small GTPase Arl1 without its GEF activity. The authors also identify the critical residues for dimerization of Gea2 and association of Gea2 and Arl1. These residues are also required for Arl1 to recruit the Arl1 effector Golgin Imh1 to the late Golgi membrane. This manuscript reveals the molecular events involved in the interaction of the Gea2-Arl1-Drs2 complex and helps to clarify the mechanisms of Arl1 regulation at late Golgi membrane trafficking. Overall, the experiments and the results are clearly presented. Additional suggestions to strengthen this study are listed below.

Important points to address:

(1) In Figure 5, the authors show that the Arl1 L69A or Y78L single mutation decreases Arl1-Gea2 association in cells. Whether the double mutation of L69A and Y78L would completely abolish the Arl1-Gea2 interaction? Since Arl1 deletion attenuates Gea2-Drs2 interaction (Tsai, et al. 2013, PNAS), the question is whether Arl1-L69A-Y78L double mutation would impair Arl1-Drs2 interaction, Gea2-Drs2 interaction, and Drs2 function?

(2) In Supplementary Figure 11, the authors show that Gea2 dimerization is required for Arl1 to recruit Imh1 to the Golgi membrane, whereas expression of the dimerization-defective Gea2 mutant (K124A L128A) results in mCherry-Imh1 diffusing. However, the authors did not explain how Gea2 dimerization determines Arl1 function. The authors should test whether the Gea2 K124A L128A mutation affects direct interaction in the Gea2-Arl1-Drs2 complex and whether the Gea2 K124A L128A mutation affects Drs2 function using the papuamide B sensitivity assay.

(3) Similar to Supplemental Figure 11, the authors should use mCherry Imh1 and co-expression of mNG fusion Arl1 in the arl1-deleted mutant to determine the cellular localization of Imh1 in Figure 6. The imaging results in Figure 6 and Supplemental Figure 11 should also be quantified and considered together with the late Golgi marker.

(4) In the Gea2-Arl1-Drs2 complex, the interaction between these proteins contributes to the stabilization of the complex. Gea2 interacts directly with Arl1 in vitro without the existence of Drs2. However, deletion of *drs2* in cells impairs the Gea2-Arl1 interaction (Tsai, et al. 2013, PNAS). In this manuscript, the authors also show the direct interaction domains between Arl1 and the Gea2 dimer without affecting the Gea2 dimer structure. However, the role of Drs2 in this complex interaction remains to be elucidated. Can the authors provide an explanation or simulated results to better elucidate the relationship between these protein interactions?

(5) In Figure 7, the authors propose a function of Arl1 that supports the association of Gea2 with the Golgi membrane. Is the Arl1-Gea2 interaction required for Gea2 membrane recruitment? The authors should examine whether an Arl1 L69A Y78L double deletion (Gea2-binding deficiency mutant) impairs Gea2 Golgi localization.

Minor points that should be addressed:

(1) The co-immunoprecipitation data in Figure 5 and Supplemental Figure 10 should include all input from the entire set of experiments. In addition, the results should be quantified with three biological replicates.

(2) In Figure 5 and Supplementary Figure 10, the authors use the co-immunoprecipitation assay to show the interaction of Gea2 dimer or Gea2-Arl1 in cells. However, the authors did not provide evidence for a direct interaction between these proteins. The authors should perform an in vitro binding assay to further substantiate their results.

(3) In Supplementary Figure 1a, the recombinant Gea2 protein was eluted by size exclusion chromatography in a fraction greater than 669 kDa. This result suggests that native Gea2 forms a tetramer. How do the authors explain the dimeric structure of Gea2 in cryo-EM, which is not consistent with the size exclusion chromatography result? The authors should use analytical ultracentrifugation to detect the dimeric structure of Gea2.

(4) There is not enough detail provided in the methods and figure legends.

Reviewer #2 (Remarks to the Author):

This manuscript is an important and interesting study revealing the structure of the Arf family guanine nucleotide exchange factor Gea2, alone and bound to Arl1. The Gea2-Arl1 complex binds to and activates the phosphatidylserine flippase Drs2, which is required for vesicle budding from the TGN. Arl1 also recruits the golgin Imh1 to the late Golgi. Interestingly, the authors show that a dimer of Gea2 can bind two molecules of Arl1, in equivalent positions on the DCB domain of each monomer, and that Gea2 dimerization is required for Imh1 recruitment to membranes. The fact that Arl1 can bind the Gea2 dimer is a surprise, because a previous structure of an Arl1-Arf GEF complex, that of Arl1 bound to the N-terminus of the related BIG1 Arf GEF, showed that Arl1 bound to the surface of the BIG1 dimerization domain (DCB) in a way that would prevent dimerization. Here, the authors show that Arl1 binds to the opposite side of the DCB domain, allowing dimerization and Arl1 binding at the same time. Although Gea2 dimerization is required for Imh1 binding to membranes, dimerization is not required for viability of yeast, or for Gea2 localization to the Golgi in yeast, as demonstrated by mutagenesis experiments. The authors also provide in vivo support for the Arl1-Gea2 interface, by mutating the residues involved and showing that interaction is abolished in co-immunoprecipitation experiments. The Gea2 dimerization interaction site is similarly tested by mutagenesis and co-immunoprecipitation experiments in yeast. This study is excellent, with exceptionally high-quality data. I have only a few minor comments that need to be addressed.

Detailed comments:

1. Page 2 lines 11-13. This sentence is not correct. Arf1 also recruits a coiled-coil tether, GMAP-210, to the cis-Golgi (see references below). Arl1 recruits a number of different coiled-coil tethers, mainly to the trans-Golgi and TGN. The difference, therefore, is that Arf1 recruits a coiled-coil tether to the early Golgi, whereas Arl1 recruits coiled-coil tethers to the late Golgi.

References:

- The GTPase Arf1p and the ER to Golgi cargo receptor Erv14p cooperate to recruit the golgin Rud3p to the cis-Golgi.

Gillingham AK, Tong AH, Boone C, Munro S. J Cell Biol. 2004 Oct 25;167(2):281-92. doi: 10.1083/jcb.200407088. PMID: 15504911

- Asymmetric tethering of flat and curved lipid membranes by a golgin.

Drin G, Morello V, Casella JF, Gounon P, Antonny B. Science. 2008 May 2;320(5876):670-3. doi: 10.1126/science.1155821. PMID: 18451304

2. Page 3, lines 21-22. The following sentence must be changed:

« ...the removal of the N-terminal 17-residue amphipathic α -helix renders the protein in a membrane-associated conformation yet it remains soluble without the membrane environment (Ref. 19). »

The cited reference (Goldberg et al. 1999) is a crystallographic study carried out without membranes. Removal of the N-terminal amphipathic α -helices (AHs) of Arf family proteins renders them soluble, with or without membranes present. Goldberg et al. 1999 showed that delta-AH Arf1 bound to the nonhydrolyzable GTP analog GppNHp adopts the GTP-bound conformation. Perhaps the authors mean that the delta-AH versions of Arf family proteins can adopt the GTP-bound conformation, which is membrane-bound when the AH is present.

3. Page 3 line 36. The phrase “The domain boundary corrects the previous sequence-based definition ...” should be changed to “The domain boundary corrects the previous sequence homology-based definition ...”.

Note that the HUS and HDS1, 2 and 3 domains were defined based on sequence homology. HUS stands for “Homology Upstream of Sec7” and HDS stands for “Homology Downstream of Sec7”. The structural domains the authors have identified contain these sequence homology domains, and also flanking sequence with no obvious homology. The sequence homology and structural domains therefore overlap, but are not identical. For simplicity, it is fine to keep the sequence homology domain names for the structural domains, as the former are contained within the latter. It might be useful to make this point on nomenclature in the manuscript, but I leave that to the authors, and don't insist on adding this discussion.

4. The manuscript should be read over and corrected for proper English usage. There are a number of incorrect formulations. I cite a few here, but this list is not exhaustive:

- Page 3 line 5 “...the previously reported crystal structures of the homolog complex...” should be “...the previously reported crystal structures of the homologous complex...”.

- Page 7, lines 2-3. “ ...Gea2 complex can also interact with the GRIP domain of Imh1, because the one Arl1 is unable to interact with both Gea2 and Imh1 simultaneously using the same binding surface.” should be changed to “ ...Gea2 complex can also interact with the GRIP domain of Imh1, because Arl1 is unable to interact with both Gea2 and Imh1 simultaneously due to the fact that the same binding surface is involved in both interactions.”

- Page 7 line 20 “At first glimpse, one would assume that the concave surface of the Arl1–Gea2 complex may bind...” is not correct, the phrase is “At first glance, one would assume that the concave surface of the Arl1–Gea2 complex may bind...”

Reviewer #3 (Remarks to the Author):

This manuscript by Duan, Jain & colleagues describes the cryo-EM structure of the Gea2 Arf1-GEF, which forms a dimer, and its complex with Arl1 combined with a mutational analysis of interfaces. The structure of the Gea2 dimer has been reported before, as well as a crystal structure of Arl1 bound to a DCB domain fragment of the Gea2 homolog BIG1. The surprising finding of this structure is that in the full-length cryo-EM structure, Arl1 binds at the opposite side of the DCB domain as in the truncated crystal structure.

The authors suggest that the Arl1 binding observed in the crystal structures is an artefact of truncating BIG1, which is an obvious explanation. However, no biochemical data is presented that supports this hypothesis. Instead, none of the mutations introduced in Gea2 in the cryo-EM interface disrupted Arl1 binding. Also, mutations that abolish Gea2 dimerization had no functional effect in yeast cells. It is important to note that the crystal structures were obtained of complexes that were produced by co-expression, while for the cryo-EM structure, dimeric Gea2 and Arl1 were purified separately and mixed. Thus, alternatively, the Gea2 dimer may be an artefact. With the available data this cannot be resolved.

For the considerations on membrane association in Fig 7, the Arl1-Gea2-Arf1 complex should be modeled and investigated.

Also, this reviewer felt that the title is misleading. The manuscript does not provide any data that “an Arl1-Arf1GEF complex [is] required for Golgi recruitment of a GRIP-domain golgin”. If anything, the structure and mutational analysis in Fig. 6 show that the same interface of Arl1 binds to Gea2 and Imh1, thus likely not at the same time. This would rather argue that the two functions of Arl1 are independent.

Reviewer #1 (Remarks to the Author):

In this manuscript, the authors provide structure-based evidence to gain insight into the interaction between the late Golgi Arf-GEF Gea2 and the Arf-like small GTPase Arl1. The authors perform cryo-EM showing that Arl1 binds to the outside surface of the dimerization region of Gea2 (DCB domain) without affecting the dimer structure of Gea2. Recently, Fromme lab reported that Arf1 binds to the Sec7 domain of Gea2 (GEF domain) to cause a conformational change of Gea2 and subsequent activation (Muccini et al. 2022, Cell Reports 40:111282). These results support the idea of how the Arf-GEF Gea2 is involved in the function of the small GTPase Arl1 without its GEF activity. The authors also identify the critical residues for dimerization of Gea2 and association of Gea2 and Arl1. These residues are also required for Arl1 to recruit the Arl1 effector Golgin Imh1 to the late Golgi membrane. This manuscript reveals the molecular events involved in the interaction of the Gea2-Arl1-Drs2 complex and helps to clarify the mechanisms of Arl1 regulation at late Golgi membrane trafficking. Overall, the experiments and the results are clearly presented. Additional suggestions to strengthen this study are listed below.

We thank the reviewer for the positive comments.

Important points to address:

(1) In Figure 5, the authors show that the Arl1 L69A or Y78L single mutation decreases Arl1-Gea2 association in cells. Whether the double mutation of L69A and Y78L would completely abolish the Arl1-Gea2 interaction? Since Arl1 deletion attenuates Gea2-Drs2 interaction (Tsai, et al. 2013, PNAS), the question is whether Arl1-L69A-Y78L double mutation would impair Arl1-Drs2 interaction, Gea2-Drs2 interaction, and Drs2 function?

Thanks for the good suggestion. We have now created the Arl1 L69A, Y78L double mutant and performed a Co-IP assay for Arl1-Gea2. Our data suggest that the Arl1 double mutant completely disrupted the Arl1-Gea2 interaction (Fig. 5). We appreciate the reviewer's concern about whether it will affect the Gea2-Drs2 interaction, but our current study is mainly focused on Gea2-Arl1 and in the future we would like to get a ternary complex of Gea2-Arl1-Drs2. Testing the Drs2 function would be out of scope for this manuscript.

(2) In Supplementary Figure 11, the authors show that Gea2 dimerization is required for Arl1 to recruit Imh1 to the Golgi membrane, whereas expression of the dimerization-defective Gea2 mutant (K124A L128A) results in mCherry-Imh1 diffusing. However, the authors did not explain how Gea2 dimerization determines Arl1 function. The authors should test whether the Gea2 K124A L128A mutation affects direct interaction in the Gea2-Arl1-Drs2 complex and whether the Gea2 K124A L128A mutation affects Drs2 function using the papuamide B sensitivity assay.

We have performed the papuamide A sensitivity assay for *gea2Δ*, *arl1Δ*, *drs2Δ* and Gea2 dimerization defective mutant (*Gea2 K124A, L128A*) strains. Our data suggest that *gea2* and *arl1* mutants are not as sensitive to papuamide A as *drs2* mutants which are hypersensitive to papuamide A (see Figure attached to the right). We are not including this result in the current manuscript as we would like to further explore the Gea2-Arl1-Drs2 ternary complex. It is also possible that *gea2* mutants will be sensitive to papuamide A at higher concentration but currently we have very limited supplies of papuamide A.

(3) Similar to Supplemental Figure 11, the authors should use mCherry Imh1 and co-expression of mNG fusion Arl1

in the *arl1*-deleted mutant to determine the cellular localization of Imh1 in Figure 6. The imaging results in Figure 6 and Supplemental Figure 11 should also be quantified and considered together with the late Golgi marker.

To address the reviewer's comment, we have now tested the localization of mCherry-Imh1 with a late Golgi marker Sec7-GFP and quantified the colocalization of Imh1-Sec7 (Fig. 6). Also, as per the reviewer's suggestion, we examined and quantified the colocalization of mNeonGreen tagged Arl1 or Gea2 constructs with a late Golgi marker Sec7-mCherry (Fig. 6, Supplementary Fig. 12). Note that the numbering for supplementary figures has increased by one.

(4) In the Gea2-Arl1-Drs2 complex, the interaction between these proteins contributes to the stabilization of the complex. Gea2 interacts directly with Arl1 *in vitro* without the existence of Drs2. However, deletion of *drs2* in cells impairs the Gea2-Arl1 interaction (Tsai, et al. 2013, PNAS). In this manuscript, the authors also show the direct interaction domains between Arl1 and the Gea2 dimer without affecting the Gea2 dimer structure. However, the role of Drs2 in this complex interaction remains to be elucidated. Can the authors provide an explanation or simulated results to better elucidate the relationship between these protein interactions?

Indeed, the role of Drs2 in this complex interaction remains to be elucidated. We will address the detailed interactions in a future work, but this is beyond the scope of the current study.

(5) In Figure 7, the authors propose a function of Arl1 that supports the association of Gea2 with the Golgi membrane. Is the Arl1-Gea2 interaction required for Gea2 membrane recruitment? The authors should examine whether an Arl1 L69A Y78L double deletion (Gea2-binding deficiency mutant) impairs Gea2 Golgi localization.

Our data and also the data previously shown by Tsai et al., 2013 suggest that there is no change in Gea2 localization in *arl1Δ* mutant. We observed no significant difference in the Gea2-mNG localization in *arl1Δ* cells expressing the Arl1 L69A, Y78L double mutant (Supplementary Fig. 12c, f).

Minor points that should be addressed:

(1) The co-immunoprecipitation data in Figure 5 and Supplemental Figure 10 should include all input from the entire set of experiments. In addition, the results should be quantified with three biological replicates.

We have provided all the input for the entire set of experiments and quantified the co-immunoprecipitation data in revised figures.

(2) In Figure 5 and Supplementary Figure 10, the authors use the co-immunoprecipitation assay to show the interaction of Gea2 dimer or Gea2-Arl1 in cells. However, the authors did not provide evidence for a direct interaction between these proteins. The authors should perform an *in vitro* binding assay to further substantiate their results.

Thanks for the good suggestion. We have now performed gel filtration chromatography to demonstrate *in vitro* binding between Gea2 and Arl1 (Supplementary Fig. 1c-d).

(3) In Supplementary Figure 1a, the recombinant Gea2 protein was eluted by size exclusion chromatography in a fraction greater than 669 kDa. This result suggests that native Gea2 forms a tetramer. How do the authors explain the dimeric structure of Gea2 in cryo-EM, which is not consistent with the size exclusion chromatography result? The authors should use analytical ultracentrifugation to detect the dimeric structure of Gea2.

Elution volume in a size exclusion chromatography profile is best interpreted for globular proteins, i.e., proteins that has a spherical shape upon folding. Given the Gea2 dimer is highly elongated, its elution volume is expected to be substantially larger than a highly compact (globular) protein of the same size. Nonetheless, there is only one

peak for the target protein indicating that Gea2 exists in a single oligomeric state. Our cryo-EM data further revealed that this single oligomeric state of Gea2 is a dimer. Indeed, out of over four million Gea2 molecule images captured by EM, they were all dimers, and none was a tetramer. While we appreciate the suggestion of AUC, we wish to note that cryo-EM is a “solution” method in which the sample is directly immobilized (rapidly frozen) in the purification buffer. So, arguably, cryo-EM provides the most definitive characterization of the Gea2 oligomeric state, which is more accurate than other biophysical methods such as the traditional AUC and the more recently developed mass photometry.

To experimentally address the reviewer’s concern, we have performed a mass photometry and determined the mass of Gea2 in solution to be (334±16) kDa (dimers) for the predominant peak and (163±16) kDa (monomers) for the minor peak (Supplementary Fig. 2). But we did not observe tetramers.

(4) There is not enough detail provided in the methods and figure legends.

We have added more descriptions in Methods and figure legends in the revised manuscript.

Reviewer #2 (Remarks to the Author):

This manuscript is an important and interesting study revealing the structure of the Arf family guanine nucleotide exchange factor Gea2, alone and bound to Arl1. The Gea2-Arl1 complex binds to and activates the phosphatidylserine flippase Drs2, which is required for vesicle budding from the TGN. Arl1 also recruits the golgin Imh1 to the late Golgi. Interestingly, the authors show that a dimer of Gea2 can bind two molecules of Arl1, in equivalent positions on the DCB domain of each monomer, and that Gea2 dimerization is required for Imh1 recruitment to membranes. The fact that Arl1 can bind the Gea2 dimer is a surprise, because a previous structure of an Arl1-Arf GEF complex, that of Arl1 bound to the N-terminus of the related BIG1 Arf GEF, showed that Arl1 bound to the surface of the BIG1 dimerization domain (DCB) in a way that would prevent dimerization. Here, the authors show that Arl1 binds to the opposite side of the DCB domain, allowing dimerization and Arl1 binding at the same time. Although Gea2 dimerization is required for Imh1 binding to membranes, dimerization is not required for viability of yeast, or for Gea2 localization to the Golgi in yeast, as demonstrated by mutagenesis experiments. The authors also provide in vivo support for the Arl1-Gea2 interface, by mutating the residues involved and showing that interaction is abolished in co-immunoprecipitation experiments. The Gea2 dimerization interaction site is similarly tested by mutagenesis and co-immunoprecipitation experiments in yeast. This study is excellent, with exceptionally high-quality data. I have only a few minor comments that need to be addressed.

We very much appreciate the reviewer’s positive comment on our work.

Detailed comments:

1. Page 2 lines 11-13. This sentence is not correct. Arf1 also recruits a coiled-coil tether, GMAP-210, to the cis-Golgi (see references below). Arl1 recruits a number of different coiled-coil tethers, mainly to the trans-Golgi and TGN. The difference, therefore, is that Arf1 recruits a coiled-coil tether to the early Golgi, whereas Arl1 recruits coiled-coil tethers to the late Golgi.

References:

- The GTPase Arf1p and the ER to Golgi cargo receptor Erv14p cooperate to recruit the golgin Rud3p to the cis-Golgi. Gillingham AK, Tong AH, Boone C, Munro S. *J Cell Biol.* 2004 Oct 25;167(2):281-92. doi: 10.1083/jcb.200407088. PMID: 15504911
- Asymmetric tethering of flat and curved lipid membranes by a golgin. Drin G, Morello V, Casella JF, Gounon P, Antony B. *Science.* 2008 May 2;320(5876):670-3. doi: 10.1126/science.1155821. PMID: 18451304

We thank this reviewer for catching the error. We have added the coiled-coil tether recruitment to the cis-Golgi and cited the references in the revised text.

2. Page 3, lines 21-22. The following sentence must be changed: « ...the removal of the N-terminal 17-residue amphipathic α -helix renders the protein in a membrane-associated conformation yet it remains soluble without the membrane environment (Ref. 19). »

The cited reference (Goldberg et al. 1999) is a crystallographic study carried out without membranes. Removal of the N-terminal amphipathic α -helices (AHs) of Arf family proteins renders them soluble, with or without membranes present. Goldberg et al. 1999 showed that delta-AH Arf1 bound to the nonhydrolyzable GTP analog GppNHp adopts the GTP-bound conformation. Perhaps the authors mean that the delta-AH versions of Arf family proteins can adopt the GTP-bound conformation, which is membrane-bound when the AH is present.

Again, we thank the reviewer for catching the erroneous statement. We have revised per the reviewer's advice.

3. Page 3 line 36. The phrase "The domain boundary corrects the previous sequence-based definition ..." should be changed to "The domain boundary corrects the previous sequence homology-based definition ...".

Note that the HUS and HDS1, 2 and 3 domains were defined based on sequence homology. HUS stands for "Homology Upstream of Sec7" and HDS stands for "Homology Downstream of Sec7". The structural domains the authors have identified contain these sequence homology domains, and also flanking sequence with no obvious homology. The sequence homology and structural domains therefore overlap, but are not identical. For simplicity, it is fine to keep the sequence homology domain names for the structural domains, as the former are contained within the latter. It might be useful to make this point on nomenclature in the manuscript, but I leave that to the authors, and don't insist on adding this discussion.

We have corrected the sentence. The domain nomenclature is explained in Figure 1 legend, so we did not repeat it here in order not to break the flow of the thought.

4. The manuscript should be read over and corrected for proper English usage. There are a number of incorrect formulations. I cite a few here, but this list is not exhaustive:

- Page 3 line 5 "...the previously reported crystal structures of the homolog complex..." should be "...the previously reported crystal structures of the homologous complex..."

- Page 7, lines 2-3. "...Gea2 complex can also interact with the GRIP domain of Imh1, because the one Arl1 is unable to interact with both Gea2 and Imh1 simultaneously using the same binding surface." should be changed to "...Gea2 complex can also interact with the GRIP domain of Imh1, because Arl1 is unable to interact with both Gea2 and Imh1 simultaneously due to the fact that the same binding surface is involved in both interactions."

- Page 7 line 20 "At first glimpse, one would assume that the concave surface of the Arl1-Gea2 complex may bind..." is not correct, the phrase is "At first glance, one would assume that the concave surface of the Arl1-Gea2 complex may bind..."

Thanks for these suggestions. We have made the corrections accordingly.

Reviewer #3 (Remarks to the Author):

This manuscript by Duan, Jain & colleagues describes the cryo-EM structure of the Gea2 Arf1-GEF, which forms a dimer, and its complex with Arl1 combined with a mutational analysis of interfaces. The structure of the Gea2 dimer has been reported before, as well as a crystal structure of Arl1 bound to a DCB domain fragment of the Gea2 homolog BIG1. The surprising finding of this structure is that in the full-length cryo-EM structure, Arl1 binds at the opposite side of the DCB domain as in the truncated crystal structure.

The authors suggest that the Arl1 binding observed in the crystal structures is an artefact of truncating BIG1,

which is an obvious explanation. However, no biochemical data is presented that supports this hypothesis. Instead, none of the mutations introduced in Gea2 in the cryo-EM interface disrupted Arl1 binding. Also, mutations that abolish Gea2 dimerization had no functional effect in yeast cells. It is important to note that the crystal structures were obtained of complexes that were produced by co-expression, while for the cryo-EM structure, dimeric Gea2 and Arl1 were purified separately and mixed. Thus, alternatively, the Gea2 dimer may be an artefact. With the available data this cannot be resolved.

We appreciate the reviewer's comment. The Gea2 dimer interface involves both DCB and HUS domains and is over 4 times larger than that of the Arl1–Gea2 complex (2524 Å² vs 594 Å²). It is highly unlikely that Arl1 can compete off such strong and extensive interactions. Further, the Gea2 dimer interface involves extensive hydrophobic interactions that could unlikely be attributed to in vitro artefact. Indeed, the Gea2 dimer structure has also been observed by the Fromme lab and shown to be functional (Muccini et al. 2022, Cell Reports 40:111282). More importantly, a most recent negative staining EM study revealed that the full-length human GBF1 indeed exists as a dimer mediated by the DCB and HUS domains (PMID: 37745300).

To further address the reviewer's concern, we have added a caveat in revision – we now note that we do not rule out the possibility that BIG1 behaves differently, as BIG1 and Gea2 belong to different ArfGEF subfamilies such that the observed interface in the crystal structure is physiologically relevant.

For the considerations on membrane association in Fig 7, the Arl1-Gea2-Arf1 complex should be modeled and investigated.

Thanks for the suggestion. We have now modeled Arf1 binding to Gea2 (See the right figure). However, there is a severe steric clash between Switch 1 of Arf1 and the loop preceding the C-linker of the Sec7 domain of Gea2, suggesting the Gea2 conformation in the Arl1–Gea2 complex is incompatible with binding to Arf1. The same observations were made by Muccini et al. (2022 Cell Reports) when Arf1 was modeled to their Gea2 in the closed conformation, which is similar to our structure. Because we did not discuss Arf1 in our MS, Arf1 is not immediately relevant to our current study, and Gea2 is unlikely to bind both Arl1 and Arf1 at the same time, we did not include the modeling result in the revised manuscript. Inclusion of the modeling would require further experiment validation that is beyond the scope of our current study.

Also, this reviewer felt that the title is misleading. The manuscript does not provide any data that “an Arl1-Arf1GEF complex [is] required for Golgi recruitment of a GRIP-domain golgin”. If anything, the structure and mutational analysis in Fig. 6 show that the same interface of Arl1 binds to Gea2 and Imh1, thus likely not at the same time. This would rather argue that the two functions of Arl1 are independent.

Indeed, as the reviewer points out, these are not simultaneous events. Rather, our title highlights a cause-and-effect relationship between the Arl1-Gea2 complex and recruitment of Imh1. And this sequential relation is supported by our in vivo studies.

REVIEWER COMMENTS

Reviewer #1 (Remarks to the Author):

I appreciate that the authors have tried to address most of the reviewers' comments. However, my previous concern about whether the Arl1-L69A-Y78L double mutation would affect Arl1-Drs2 interaction, Gea2-Drs2 interaction, and Drs2 function unfortunately remains unanswered.

The authors stated that Arl1 is unable to interact with both Gea2 and Imh1 simultaneously due to the fact that the same binding surface is involved in both interactions. Therefore, it is reasonable to speculate that the GTP-bound state Arl1 likely needs to dissociate from the Gea2 dimer before it can interact with the Imh1 GRIP domain. However, the authors should provide an explanation or simulated results to better understand the relationship between these protein interactions.

It has been shown that Arl1 can remain at the late Golgi in *gea2*-deletion or *drs2*-deletion cells. In Fig. 6b, the authors show that the Golgi localizations of Arl1 single mutants L69A, Y78L and double mutant L69A, Y78L were all decreased compared to wild-type Arl1. Since Imh1 requires the GTP-bound form of Arl1 to target it to the late Golgi, it appears that these Arl1 mutants are also defective in GTP binding, resulting in their diffuse distribution in the cytoplasm. Consequently, Imh1 was diffused in the cytoplasm in yeast carrying these Arl1 mutations. Therefore, the importance of the Arl1 binding surface for Arl1-Gea2 interaction could also have implications for GTP binding of Arl1.

Reviewer #3 (Remarks to the Author):

The revised version by Duan, Jain & colleagues includes additional discussion to address the comments of the previous review.

Regarding the Arl1 interface on Gea2 EM vs. BIG X-ray, for which the most obvious explanation is also likely true, this is fine.

However, dismissing Arf1 from the final model (Fig7 – the membrane is missing in panel a in the revised version) because “Arf1 is not immediately relevant to our current study” seems a bit rash. Gea2 is after all an Arf1 GEF and in Fig 3 the authors already included a comparison of their structure to the Arf1-Gea2 catalytic intermediate. It is an obvious question to ask if Arl1, Gea2 (in the open GEF structure) and Arf1 could interact with the membrane simultaneously or to show actually show that “Gea2 is unlikely to bind both Arl1 and Arf1 at the same time”. With the data from this manuscript and the results by Muccini et al., such a model could be presented without the need of an experimental validation.

Admittedly, I am not an expert on Golgi GTPases and their effectors, but “a cause-and-effect relationship between the Arl1-Gea2 complex and recruitment of Imh1” does not become clear in the manuscript. The data show that Gea2 and Arl1 are involved in Imh1 localization (Fig 6b misses a label), but is the complex? There is literature on Arl1-Imh1 interaction with or without Gea2. And since Imh1 is an Arl1 effector, the Arl1 mutations could directly affect the interaction with Imh1 and loss of Golgi localization may be independent of Gea2. A much more detailed discussion (or additional data) is needed to maintain the claim of the title.

None of these issues are essential to warrant publication of the manuscript, but would make the paper much more useful for a general audience.

REVIEWER COMMENTS

Reviewer #1 (Remarks to the Author):

I appreciate that the authors have tried to address most of the reviewers' comments. However, my previous concern about whether the Arl1-L69A-Y78L double mutation would affect Arl1-Drs2 interaction, Gea2-Drs2 interaction, and Drs2 function unfortunately remains unanswered.

We agree with the reviewer the Drs2 interaction aspect remains unaddressed. As we noted in the manuscript and the response letter, our current focus is on the binary Arl1-Gea2 complex. We have performed an extensive series of structural and cellular studies to elucidate the Arl1 interaction with Gea2. It is very much our desire to address the Drs2 interaction by solving the ternary complex structure and interrogating their interactions. However, our preliminary investigation indicates this is not trivial and cannot be accomplished in a short time. In addition, the regulation of Drs2 activity in vivo appears to be quite complex. For example, the site where Gea2 binds Drs2 is also where the F-box protein Rcy1 binds (PMID 24272750) and is adjacent to a region that binds phosphoinositides (PMID 19898464). All of these interactions appear to regulate Drs2 and the degree of redundancy is unknown. Substantial effort is required to tease out the contribution of Arl1-Gea2 relative to these other regulators. Therefore, the Drs2 aspect must be left for future work.

The authors stated that Arl1 is unable to interact with both Gea2 and Imh1 simultaneously due to the fact that the same binding surface is involved in both interactions. Therefore, it is reasonable to speculate that the GTP-bound state Arl1 likely needs to dissociate from the Gea2 dimer before it can interact with the Imh1 GRIP domain. However, the authors should provide an explanation or simulated results to better understand the relationship between these protein interactions.

We apologize for not explaining this issue clearly. We have now added surface views (Fig. 6d) to better illustrate the fact that Imh1 binds to the same surface on Arl1 as Gea2, therefore, Imh1 and Gea2 cannot bind to Arl1 at the same time, and Arl1 has to dissociate from Gea2 before interacting with Imh1. This new panel complements Supplementary Figure 13a, which shows similar information but in more detail.

Furthermore, we have also provided an alternative and perhaps more straightforward explanation in the revised main text, that there may be a separate pool of Arl1 that recruits Imh1 to the Golgi.

It has been shown that Arl1 can remain at the late Golgi in *gea2*-deletion or *drs2*-deletion cells. In Fig. 6b, the authors show that the Golgi localizations of Arl1 single mutants L69A,

Y78L and double mutant L69A, Y78L were all decreased compared to wild-type Arl1. Since Imh1 requires the GTP-bound form of Arl1 to target it to the late Golgi, it appears that these Arl1 mutants are also defective in GTP binding, resulting in their diffuse distribution in the cytoplasm. Consequently, Imh1 was diffused in the cytoplasm in yeast carrying these Arl1 mutations. Therefore, the importance of the Arl1 binding surface for Arl1-Gea2 interaction could also have implications for GTP binding of Arl1.

Thank you for the thought-provoking comments. But both L69 and Y78 are far away from the gamma-phosphate of the bound GTP in our structure, and their mutation(s) are unlikely to directly affect GTP binding. However, the reviewer is correct that the mutations appear to increase the Arl1-GDP levels in the cell. We have revised the manuscript on page 8 to provide two possible explanations for this result. Either the Arl1 mutations perturb interaction with Syt1, the Arl1-GEF, or increase its accessibility to Gcs1, the Arl1-GAP.

Reviewer #3 (Remarks to the Author):

The revised version by Duan, Jain & colleagues includes additional discussion to address the comments of the previous review.

Regarding the Arl1 interface on Gea2 EM vs. BIG X-ray, for which the most obvious explanation is also likely true, this is fine.

Thank you for raising the issue in the first place. After submission of our previous revision, a paper appeared in BioRxiv (<https://www.biorxiv.org/content/10.1101/2023.11.22.568272v1>), which showed that a BIG1 close homolog dimerized via a unique HDS4 domain, a domain absent in Gea2. This new work suggests that BIG1 dimerizes via the unique HDS4 and that the BIG1 dimer interface is different from the Gea2 dimer interface, clearly supporting the notion that the co-crystal structure reported by the Munro lab is most likely physiologically relevant. We have revised accordingly and added a comparison of the distinct domain architectures of ARL1-BIG1 and Arl1-Gea2 (Fig. 4c-d).

However, dismissing Arf1 from the final model (Fig7 – the membrane is missing in panel a in the revised version) because “Arf1 is not immediately relevant to our current study” seems a bit rash. Gea2 is after all an Arf1 GEF and in Fig 3 the authors already included a comparison of their structure to the Arf1-Gea2 catalytic intermediate. It is an obvious question to ask if Arl1, Gea2 (in the open GEF structure) and Arf1 could interact with the membrane simultaneously or to show actually show that “Gea2 is unlikely to bind both Arl1 and Arf1 at the same time”. With the data from this manuscript and the results by Muccini et al., such a model could be presented without the need of an experimental validation.

The membrane drawing was lost during conversion from .doc to .pdf. This has been corrected. Thanks for pointing out the modeling rationale which we fully agree. We have

now modeled (1) Arf1-GDP into our Arl1-Gea2 structure and (2) Arl1-GTP into the published Arf1-Gea2 structure (Fig. 7c-d). The modeling results indicate that Gea2 may interact with Arf1-GDP and Arl1-GTP simultaneously. However, in the presence of both Arl1 and Arf1, the GEF reaction would lead to a large rotation of the Sec7 domain that inserts Arf1-NF into the membrane core. It is currently unclear if the local membrane structure may undergo a compensatory change to accommodate the structural changes in Arf1-NF, or that the enzyme complex may move away from the membrane to avoid such clash.

Admittedly, I am not an expert on Golgi GTPases and their effectors, but "a cause-and-effect relationship between the Arl1-Gea2 complex and recruitment of Imh1" does not become clear in the manuscript. The data show that Gea2 and Arl1 are involved in Imh1 localization (Fig 6b misses a label), but is the complex? There is literature on Arl1-Imh1 interaction with or without Gea2. And since Imh1 is an Arl1 effector, the Arl1 mutations could directly affect the interaction with Imh1 and loss of Golgi localization may be independent of Gea2. A much more detailed discussion (or additional data) is needed to maintain the claim of the title.

None of these issues are essential to warrant publication of the manuscript, but would make the paper much more useful for a general audience.

We have fixed the missing label. We have revised the title, to "*Structural insight into an Arl1-ArfGEF complex involved in Golgi recruitment of a GRIP-domain golgin*".

Finally, we thank the reviewer for the thoughtful comments, which significantly improved our manuscript.

REVIEWERS' COMMENTS

Reviewer #3 (Remarks to the Author):

The authors have addressed all remaining issues and I support the publication of this very interesting manuscript.

(note: "for" needs to be deleted from the title)